

# Carrollian $\mathscr{L}w_{1+\infty}$ representation from twistor space

**Laura Donnay[1⋆], Laurent Freidel[2†] and Yannick Herfray[3‡]**

**1** INFN, Sezione di Trieste, Via Valerio 2, 34127, Italy
**2** Perimeter Institute for Theoretical Physics, 31 Caroline St. N.,
Waterloo ON, Canada, N2L 2Y5
**3** Institut Denis Poisson UMR 7013, Université de Tours,
Parc de Grandmont, 37200 Tours, France

⋆ ldonnay@sissa.it , † lfreidel@perimeterinstitute.ca , ‡ yannick.herfray@univ-tours.fr

## Abstract

We construct an explicit realization of the action of the $\mathscr{L}w_{1+\infty}$ loop algebra on fields at null infinity. This action is directly derived by Penrose transform of the geometrical action of $\mathscr{L}w_{1+\infty}$ symmetries in twistor space, ensuring that it forms a representation of the algebra. Finally, we show that this action coincides with the canonical action of $\mathscr{L}w_{1+\infty}$ Noether charges on the asymptotic phase space.

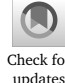

# 1 Introduction

The existence of an infinite hierarchy of conservation laws associated to every complexified self-dual Einstein manifold with zero cosmological constant can be traced back to the work of Plebanski and Boyer [1,2]. These conservation laws are related to the action of singular diffeomorphisms in twistor space, which act as symmetries and whose infinitesimal action is the $\mathscr{L}w_{1+\infty}$ algebra. The singular nature of these diffeomorphisms means that they can change the complex structure which, by Penrose's nonlinear graviton construction [3,4], amounts to deforming a self-dual spacetime into another. In this sense the $\mathscr{L}w_{1+\infty}$ algebra acts as symmetry on the space of self-dual Einstein spacetimes. In fact, several constructions in twistor theory have been making use of singular transformations as methods for generating new solutions [5–13].

The $\mathscr{L}w_{1+\infty}$ algebra has recently made a dramatic comeback in the context of the celestial holography program, whose goal is to encode quantum gravity in asymptotically flat spacetimes in terms of a theory living on the celestial sphere. Celestial amplitudes, namely scattering amplitudes recast in a conformal primary basis on the celestial sphere (see [14–17] for reviews), exhibit an infinite tower of 'conformally soft' graviton theorems, which appear when the conformal dimension of the external graviton takes certain integer values [18–21]. It was shown that, in the positive helicity sector, the infinite collection of soft graviton currents can be organized into the loop algebra of the wedge algebra of $w_{1+\infty}$ [22] (see also [23–38] for related works). In other words, $\mathscr{L}w_{1+\infty}$ provides a symmetry organizing principle for the soft sector of celestial CFTs.

The explicit relation between soft theorems and the $\mathscr{L}w_{1+\infty}$ symmetries of twistor space was realized by T. Adamo, L. Mason and A. Sharma in [27]. The main new ingredient was the use of a new twistor sigma model developed in the last years [39–42]. This new model is based on the theory of asymptotic twistor spaces [43,44], which are closely related to Newman's $\mathcal{H}$-spaces [45,46]: These are particular realisations of the nonlinear graviton construction where the deformation of twistor space is parameterized by the gravitational data (i.e. the shear) at null infinity. This point of view, together with their sigma model which permits to probe gravitational amplitudes beyond the self-dual sector, allowed these authors to establish a close connection between scattering amplitudes and the action of the $\mathscr{L}w_{1+\infty}$ algebra. In particular, their work made it very clear that in principle the algebra acts on gravitational data. However, as often in twistor theory, the construction is somewhat implicit and the exact realisation of this action was left aside. The main result of our article is that, in this particular instance, one can be particularly explicit about the action of the symmetry and find a closed expression for the action suggested by the work [27]. The second result is that the action can be extended to act on the asymptotic data of any spin.

On another front, while the $w_{1+\infty}$ structure was explicitly related to the celestial operator product expansions of soft gravitons, it was not clear how it could be seen to emerge from a gravitational phase space point of view. To remedy this situation, the works [47,48] proposed a construction of charges associated with a higher-spin tower of symmetries. One of the key properties of these higher-spin charges is that they satisfy a set of recursion relations which, once truncated to quadratic order in the fields, is equivalent to the tower of conformally soft symmetries. This allowed the authors of [48] to provide a canonical realization of the $\mathscr{L}w_{1+\infty}$ algebra on the gravitational phase space from the bracket of (a renormalized version of) these higher-spin charges. In [33,49] it was further shown that these canonical charges form a representation of a shifted Schouten-Nijenhuis algebra, which reduces to $\mathscr{L}w_{1+\infty}$ under certain holomorphicity conditions of the transformation parameters. We will in fact be able to show that this canonical realization precisely coincides with the twistor action.

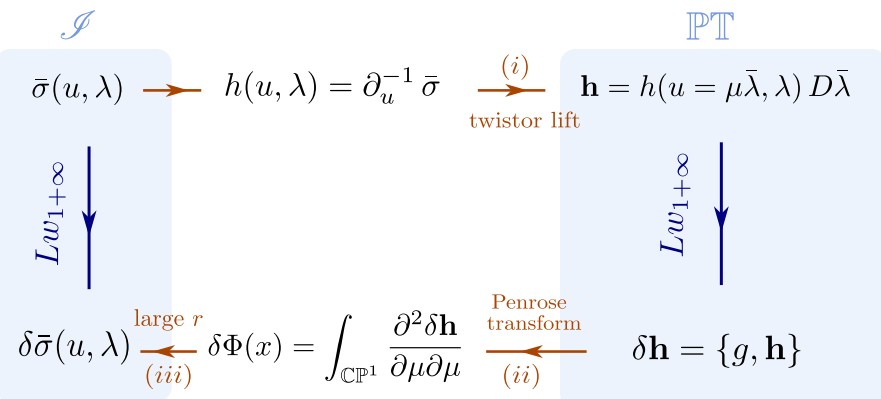

Figure 1: Schematic view of the different steps. One starts at the upper left corner with the shear $\bar{\sigma}$ at $\mathscr{I}$ (of coordinates $u, \lambda$) and constructs its uplift in twistor space, $\mathbf{h} \in \mathbb{PT}$. The action of the loop algebra $\mathscr{L}w_{1+\infty}$ of generators $g$ is linear on twistor space and renders $\delta\mathbf{h}$. A Penrose transform allows to obtain the transformed bulk field $\delta\Phi$, from where a stationary phase space approximation leads to the explicit action of $w_{1+\infty}$ symmetries on the asymptotic shear, $\delta\bar{\sigma}$.

The goal of this work is therefore to provide a direct derivation of the representation[1] of $\mathscr{L}w_{1+\infty}$ symmetries on fields living at null infinity ($\mathscr{I}$) from its twistor action. We will refer to objects intrinsically living at $\mathscr{I}$ as 'Carrollian fields', following a recent nomenclature (see [50–52] and references therein for a Carrollian perspective and see [38, 40] for recent works connecting the Carrollian and twistor perspective). Our derivation will consist of a direct computation where we 'bring down to $\mathscr{I}$' the action of $\mathscr{L}w_{1+\infty}$ symmetries in twistor space. More precisely, we will follow the sequence of steps which are outlined in Fig. 1. This intricate journey will eventually render a remarkably simple expression for the action of $\mathscr{L}w_{1+\infty}$ on the shear which the reader can find in Proposition 1. What is more, it will automatically ensure that the action forms a representation of the algebra, see Proposition 2. We will also show that our result matches with the canonical action of higher-spin charges found in [48], thereby unifying the aforementioned results on the appearance of $w_{1+\infty}$ symmetries from twistor space, celestial CFT, Carrollian and gravitational phase space points of view.

This paper is organized as follows. We start in section 2 with a presentation of conventions and the key relationships that allow us to go from twistor representatives to Carrollian fields at null infinity and back. In section 3, we present our main results, namely the explicit realization of the action of $\mathscr{L}w_{1+\infty}$ symmetries on the gravitational shear, and that the latter forms a representation of the algebra. We then show in section 4 that our expressions derived from twistor space coincide with the canonical action that was previously obtained from a gravitational phase space perspective. Section 5 contains the detailed steps of the proof of Proposition 1.

---

[1]In other terms, the linear action of $\mathscr{L}w_{1+\infty}$ on linearized fields at $\mathscr{I}$.

$$\bar{\sigma}(u,\lambda) \longrightarrow h(u,\lambda) = \partial_u^{-1}\bar{\sigma} \xrightarrow[\text{twistor lift}]{(i)} \mathbf{h} = h(u = \mu\bar{\lambda},\lambda)\, D\bar{\lambda}$$

$$\Phi(x) = \int_{\mathbb{CP}^1} \frac{\partial^2 \mathbf{h}}{\partial\mu\partial\mu}$$

Figure 2: Schematic representation of the circular journey (here depicted for the case of positive helicity) with the identification of $\bar{\sigma}$, $\mathbf{h}$ and $\Phi$ and their relationships.

## 2 From null infinity to twistor space and back

In this section, we set up our notations and detail the journey of a Carrollian field of helicity $\pm 2$ at $\mathscr{I}$ to its uplift to twistor space, and back; see Fig. 2. The first step consists in mapping the shears $\bar{\sigma}$ (resp. $\sigma$) to their twistor representatives $\mathbf{h}$ (resp. $\widetilde{\mathbf{h}}$). Applying the corresponding Penrose transform renders a field in the bulk $\Phi(x)$, from which one can recover the initial field at $\mathscr{I}$ from an asymptotic (large $r$) expansion. The consistency of this trivial journey will ensure that our conventions are consistent and prepare the ground for the action of the $\mathscr{L}w_{1+\infty}$ algebra.

### 2.1 Bondi coordinates and spinors

Our conventions for the contractions of spinors with the Levi-Civita symbol $\epsilon^{\alpha\beta}$, with $\epsilon^{01} = 1$, are $\langle ab \rangle = \epsilon^{\alpha\beta}a_\beta b_\alpha = a^\alpha b_\alpha$, $[\tilde{a}\tilde{b}] = \epsilon^{\dot{\alpha}\dot{\beta}}\tilde{a}_{\dot{\beta}}\tilde{b}_{\dot{\alpha}} = \tilde{a}^{\dot{\alpha}}\tilde{b}_{\dot{\alpha}}$ and, in particular, we will make important use of the following

$$\lambda^\alpha := \begin{pmatrix} z \\ -1 \end{pmatrix}, \qquad n^\alpha := \begin{pmatrix} 1 \\ 0 \end{pmatrix}, \qquad \langle n\lambda \rangle = 1, \qquad d\lambda^\alpha = dz\, n^\alpha, \qquad D\lambda := \langle \lambda d\lambda \rangle = -dz.$$

Bondi coordinates are chosen as $(u, r, \lambda_\alpha, \bar{\lambda}_{\dot{\alpha}})$ together with a null vector $n^{\alpha\dot{\alpha}} = n^\alpha \bar{n}^{\dot{\alpha}}$

$$\lambda_\alpha = (1, z), \qquad n^{\alpha\dot{\alpha}} = \begin{pmatrix} 1 & 0 \\ 0 & 0 \end{pmatrix}, \tag{1}$$

such that Minkowski space $\mathbb{M}$ is parametrized by

$$x^{\alpha\dot{\alpha}} = u\, n^{\alpha\dot{\alpha}} + r\, \lambda^\alpha\, \bar{\lambda}^{\dot{\alpha}} \in \mathbb{M}, \tag{2}$$

with flat metric

$$dx^{\alpha\dot{\alpha}} dx_{\alpha\dot{\alpha}} = 2du\, dr - 2r^2 dz\, d\bar{z}. \tag{3}$$

We denote by $\mathcal{C}^\infty_{k,\bar{k}}(\mathscr{I})$ the space of Carrollian fields at $\mathscr{I}$ of weight $(k, \bar{k})$. They are represented interchangeably : (i) either by functions $\phi(u, z, \bar{z})$ on $\mathbb{R} \times S^2$ with the prescribed transformation law (see e.g. [51, 52])

$$\delta_\xi \phi(u, z, \bar{z}) = \left(\mathcal{T} + \frac{u}{2}(\partial\mathcal{Y} + \bar{\partial}\bar{\mathcal{Y}})\right)\partial_u\phi(u, z, \bar{z}) + \left(\mathcal{Y}\partial + \bar{\mathcal{Y}}\bar{\partial} + k\partial\mathcal{Y} + \bar{k}\bar{\partial}\bar{\mathcal{Y}}\right)\phi(u, z, \bar{z}), \tag{4}$$

under an element[2] $\xi = \left(\mathcal{T} + \frac{u}{2}\left(\partial \mathcal{Y} + \bar{\partial}\bar{\mathcal{Y}}\right)\right)\partial_u + \mathcal{Y}\partial + \bar{\mathcal{Y}}\bar{\partial}$ of the (extended) BMS algebra [53], or by (ii) functions $\phi(u, \lambda_\alpha, \bar{\lambda}_{\dot{\alpha}})$ on[3] $\mathbb{R} \times \mathbb{C}^2$ with the following homogeneity, [44]

$$\phi\left(|b|^2 u, b\lambda_\alpha, \bar{b}\bar{\lambda}_{\dot{\alpha}}\right) = b^{-2k}\bar{b}^{-2\bar{k}}\phi\left(u, \lambda_\alpha, \bar{\lambda}_{\dot{\alpha}}\right), \tag{5}$$

under multiplication by a non zero complex number $b \in \mathbb{C}$. These are two different realisations of the same Carrollian field. The identification is explicitly given by the fact that they define the same tensorial field $\Phi \in (\Omega^{(1,0)})^k \otimes (\Omega^{(0,1)})^{\bar{k}}$ on $\mathscr{I}$,

$$\Phi = \phi(u, z, \bar{z})(-dz)^k(-d\bar{z})^{\bar{k}} = \phi(u, \lambda_\alpha, \bar{\lambda}_{\dot{\alpha}})(D\lambda)^k(\bar{D}\bar{\lambda})^{\bar{k}}. \tag{6}$$

The null momenta $p^{\alpha\dot{\alpha}}$ for a particle heading towards the point $\zeta_\alpha = (1, \zeta)$, $\bar{\zeta}_{\dot{\alpha}} = (1, \bar{\zeta})$ on the celestial sphere will be written as

$$p^{\alpha\dot{\alpha}} = \omega q^{\alpha\dot{\alpha}}(\zeta, \bar{\zeta}) = \omega\zeta^\alpha\bar{\zeta}^{\dot{\alpha}}, \tag{7}$$

and we define the polarization tensor as

$$\epsilon^{(+)}_{\alpha\dot{\alpha}} = \frac{\iota_\alpha\bar{\zeta}_{\dot{\alpha}}}{\langle\iota\zeta\rangle}, \qquad \epsilon^{(-)}_{\alpha\dot{\alpha}} = \frac{\zeta_\alpha\bar{\iota}_{\dot{\alpha}}}{[\bar{\zeta}\bar{\iota}]}, \tag{8}$$

where $\iota^\alpha$ is an unspecified reference spinor corresponding to residual gauge freedom. From (2) we also deduce that

$$\bar{\partial} = r\lambda^\alpha\bar{n}^{\dot{\alpha}}\frac{\partial}{\partial x^{\alpha\dot{\alpha}}}, \tag{9}$$

and that the contraction with the polarization tensor is

$$\bar{\partial} \lrcorner \epsilon^{(+)} = \bar{\partial}^{\alpha\dot{\alpha}}\epsilon^{(+)}_{\alpha\dot{\alpha}} = r\frac{\langle\lambda\iota\rangle}{\langle\iota\zeta\rangle}, \qquad \bar{\partial} \lrcorner \epsilon^{(-)} = \bar{\partial}^{\alpha\dot{\alpha}}\epsilon^{(-)}_{\alpha\dot{\alpha}} = r\frac{\langle\lambda\zeta\rangle[\bar{n}\bar{\iota}]}{[\bar{\zeta}\bar{\iota}]}. \tag{10}$$

In the same vein, we note that

$$x^{\alpha\dot{\alpha}}q_{\alpha\dot{\alpha}} = u + r\langle\lambda\zeta\rangle[\bar{\lambda}\bar{\zeta}] = u + r|z - \zeta|^2. \tag{11}$$

## 2.2 A trivial journey

We now detail the round trip from $\mathscr{I}$ to twistor space and back as depicted in Fig. 2, namely we check that one can recover the asymptotic shear from its uplift to twistor space by applying a Penrose transform, followed by a stationary phase space approximation. These are coordinate invariant operations and we will work with the coordinate system (3) for concreteness. Following the notations and conventions of [51], we start with the Carrollian representative (shear)[4]

$$\bar{\sigma}(u, \lambda, \bar{\lambda}) = -\frac{i\kappa}{8\pi^2}\int_0^\infty d\omega\left(a_-(\omega, \lambda, \bar{\lambda})e^{-i\omega u} - a_+^\dagger(\omega, \lambda, \bar{\lambda})e^{i\omega u}\right), \tag{12}$$

which encodes the self-dual radiative degrees of freedom [54–57], together with its opposite helicity counterpart,

$$\sigma(u, \lambda, \bar{\lambda}) = -\frac{i\kappa}{8\pi^2}\int_0^\infty d\omega\left(a_+(\omega, \lambda, \bar{\lambda})e^{-i\omega u} - a_-^\dagger(\omega, \lambda, \bar{\lambda})e^{i\omega u}\right), \tag{13}$$

---

[2]Here and everywhere in this article $\partial$ stands for the partial derivative $\frac{\partial}{\partial z}$ while $\bar{\partial}$ stands for $\frac{\partial}{\partial \bar{z}}$. To avoid potential confusion, the Dolbeault operator on twistor space will be denoted $\bar{\mathrm{d}}$.

[3]$(u, \lambda, \bar{\lambda}) \sim (|b|^2 u, b\lambda, \bar{b}\bar{\lambda})$ then stand for homogeneous coordinates on $\mathscr{I}$.

[4]The expression below really corresponds to the projection of $C_{AB}$ along the null sphere frame in Bondi coordinates, which relates to Newman-Penrose's shear as $\bar{\sigma}^{\mathrm{NP}} = \frac{1}{2}\bar{\sigma}^{\mathrm{here}}$.

which encodes the anti-self-dual radiative degrees of freedom. Here, $\kappa = \sqrt{32\pi G}$ and $a$, $a^\dagger$ are annihilation and creation operators in momentum basis. The fields' commutation relations are given by

$$[a_\alpha(\omega, \lambda, \bar{\lambda}), a^\dagger_{\alpha'}(\omega', \lambda', \bar{\lambda}')] = 16\pi^3 \delta_{\alpha,\alpha'} \omega^{-1} \delta(\omega - \omega') \delta(z - z'),$$

$$[\sigma(u, z, \bar{z}), \bar{\sigma}(u', z', \bar{z}')] = -i\frac{\kappa^2}{4} \text{sign}(u - u') \delta(z - z').$$

(14)

### $i$) Lift to twistor space $\mathscr{I} \to \mathbb{PT}$

Let us start with the positive helicity Carrollian field (12). We introduce, following [27, 44],

$$h(u, \lambda, \bar{\lambda}) = \partial_u^{-1} \bar{\sigma}(u, \lambda, \bar{\lambda}) = \frac{\kappa}{8\pi^2} \int_0^\infty \frac{d\omega}{\omega} \left( e^{-i\omega u} a_-(\omega, \lambda, \bar{\lambda}) + e^{i\omega u} a^\dagger_+(\omega, \lambda, \bar{\lambda}) \right),$$

(15)

where $\partial_u^{-1} = \int^u du$ acts on plane waves as $\partial_u^{-1} e^{i\omega u} = \frac{1}{i\omega} e^{i\omega u}$.

The uplift to twistor space of the Carrollian representative (12) then is

$$\mathbf{h}(Z^A, \bar{Z}^A) = h\left(\mu^{\dot{\alpha}} \bar{\lambda}_{\dot{\alpha}}, \lambda, \bar{\lambda}\right) D\bar{\lambda}$$

$$= -\frac{\kappa}{8\pi^2} \int_0^\infty \frac{d\omega}{\omega} \left( e^{-i\omega \mu^{\dot{\alpha}} \bar{\lambda}_{\dot{\alpha}}} a_-(\omega, \lambda, \bar{\lambda}) + e^{i\omega \mu^{\dot{\alpha}} \bar{\lambda}_{\dot{\alpha}}} a^\dagger_+(\omega, \lambda, \bar{\lambda}) \right) d\bar{z},$$

(16)

where $D\bar{\lambda} := [\bar{\lambda} d\bar{\lambda}]$ and twistor coordinates are $Z^A = \left(\mu^{\dot{\alpha}}, \lambda_\alpha\right) \in \mathbb{C}^4$. The weights of $\bar{\sigma}$ ensures that $\mathbf{h}$ is homogeneous of degree $(2, 0)$ in the twistor coordinates $\left(Z^A, \bar{Z}^A\right)$. Indeed, $\bar{\sigma}$ has Carrollian weights $(k, \bar{k}) = (-\frac{1}{2}, \frac{3}{2})$ i.e.

$$\bar{\sigma}(|b|^2 u, b\lambda_\alpha, \bar{b}\bar{\lambda}_{\dot{\alpha}}) = b^{-2k} \bar{b}^{-2\bar{k}} \bar{\sigma}(u, \lambda_\alpha, \bar{\lambda}_{\dot{\alpha}}) = \frac{b}{\bar{b}^3} \bar{\sigma}(u, \lambda_\alpha, \bar{\lambda}_{\dot{\alpha}}),$$

(17)

for any non-vanishing complex number $b$, and hence

$$\mathbf{h}(bZ^A, \bar{b}\bar{Z}^A) = b^2 \mathbf{h}(Z^A, \bar{Z}^A).$$

(18)

One can also check that the twistor representative $\mathbf{h} \in \Omega^{0,1}(\mathbb{PT}, \mathcal{O}(2))$ is holomorphic in twistor space, $\bar{\mathrm{d}}\mathbf{h} = 0$, where $\bar{\mathrm{d}} := \mathrm{d}\bar{Z}^A \frac{\partial}{\partial \bar{Z}^A}$.

For the negative helicity field (13) of Carrollian weights $(\frac{3}{2}, -\frac{1}{2})$, we introduce instead

$$\tilde{h}(u, \lambda, \bar{\lambda}) := \partial_u^3 \sigma(u, \lambda, \bar{\lambda}) = \frac{\kappa}{8\pi^2} \int_0^\infty d\omega\, \omega^3 \left( e^{-i\omega u} a_+(\omega, \lambda, \bar{\lambda}) + e^{i\omega u} a^\dagger_-(\omega, \lambda, \bar{\lambda}) \right),$$

(19)

and the following uplift to twistor space

$$\widetilde{\mathbf{h}}(Z^A, \bar{Z}^A) = \tilde{h}\left(\mu^{\dot{\alpha}} \bar{\lambda}_{\dot{\alpha}}, \lambda, \bar{\lambda}\right) D\bar{\lambda}$$

$$= -\frac{\kappa}{8\pi^2} \int_0^\infty d\omega\, \omega^3 \left( e^{-i\omega \mu^{\dot{\alpha}} \bar{\lambda}_{\dot{\alpha}}} a_+(\omega, \lambda, \bar{\lambda}) + e^{i\omega \mu^{\dot{\alpha}} \bar{\lambda}_{\dot{\alpha}}} a^\dagger_-(\omega, \lambda, \bar{\lambda}) \right) d\bar{z}.$$

(20)

The weights of $\sigma$ imply that the twistor representative $\widetilde{\mathbf{h}} \in \Omega^{0,1}(\mathbb{PT}, \mathcal{O}(-6))$ is homogeneous of degree $(-6, 0)$ in the twistor coordinates $Z^A, \bar{Z}^A$. It also satisfies $\bar{\mathrm{d}}\widetilde{\mathbf{h}} = 0$.

To summarize, we have two linear maps (one for each helicity),

$$\mathbf{T}^{+2} \left| \begin{array}{ccc} \mathcal{C}^\infty_{(-\frac{1}{2}, \frac{3}{2})}(\mathscr{I}) & \to & \Omega^{0,1}(\mathbb{PT}, \mathcal{O}(2)), \\ \bar{\sigma} & \mapsto & \mathbf{h}, \end{array} \right.$$

(21)

$$\mathbf{T}^{-2} \left| \begin{array}{ccc} \mathcal{C}^\infty_{(\frac{3}{2}, -\frac{1}{2})}(\mathscr{I}) & \to & \Omega^{0,1}(\mathbb{PT}, \mathcal{O}(-6)), \\ \sigma & \mapsto & \widetilde{\mathbf{h}}, \end{array} \right.$$

lifting the shear to the corresponding twistor representatives. For a general Carrollian field $\phi(u, z, \bar{z})$ of weight $(k, \bar{k}) = \left(\frac{1-s}{2}, \frac{1+s}{2}\right)$ we have a map

$$\mathbf{T}^s \left|\begin{array}{ccc} \mathcal{C}^\infty_{(\frac{1-s}{2}, \frac{1+s}{2})}(\mathscr{I}) & \to & \Omega^{0,1}(\mathbb{PT}, \mathcal{O}(2s-2)), \\ \phi & \mapsto & \mathbf{f} = f \bar{D}\lambda, \end{array}\right. \tag{22}$$

given by $f(Z^A, \bar{Z}^A) := \left(\partial_u^{1-s}\phi\right)(u = \mu^{\dot\alpha}\bar\lambda_{\dot\alpha}, \lambda, \bar\lambda)$.

### $ii$) Penrose transform: $\mathbb{PT} \to \mathbb{M}$

Functions $f(Z^A, \bar{Z}^B)$ on $\mathbb{T}$ which are homogeneous degree $k$ in the holomorphic twistor coordinate $Z^A$ i.e.

$$f(bZ^A, \bar{b}\bar{Z}^B) = b^k f(Z^A, \bar{Z}^B), \tag{23}$$

correspond to sections of the holomorphic bundle $\mathcal{O}(k) \to \mathbb{PT}$. Twistor representatives are given by $\bar{\mathrm{d}}$ closed (but not exact) $(0,1)$-forms $\mathbf{f} \in \Omega^{0,1}(\mathbb{PT}, \mathcal{O}(2s-2))$. The Penrose transform then identifies the corresponding cohomology class with massless fields of helicity $s \in \mathbb{Z}$ (see e.g. [58–63]):

$$\left\{\text{zero rest mass fields on } \mathbb{M}_\mathbb{C} \text{ of helicity s}\right\} \simeq H^{0,1}(\mathbb{PT}, \mathcal{O}(2s-2)). \tag{24}$$

In particular, from our twistor representative $\mathbf{h} \in \Omega^{0,1}(\mathbb{PT}, \mathcal{O}(2))$, given by (16), one can recover the corresponding positive helicity 2 fields as[5]

$$\begin{aligned} \Phi_{\alpha\dot\alpha\beta\dot\beta}(x) &= \frac{1}{2\pi i} \int_{\mathbb{CP}^1} \langle \zeta d\zeta \rangle \wedge \frac{\iota_\alpha \iota_\beta}{\langle \iota\zeta \rangle^2} \frac{\partial^2 \mathbf{h}}{\partial \mu^{\dot\alpha} \partial \mu^{\dot\beta}}(\mu^{\dot\alpha} = x^{\alpha\dot\alpha}\zeta_\alpha, \zeta_\alpha) \\ &= \frac{\kappa i}{16\pi^3} \int_{\mathbb{CP}^1} \langle \zeta d\zeta \rangle \wedge [\bar\zeta d\bar\zeta] \frac{\iota_\alpha \iota_\beta \bar\zeta_{\dot\alpha} \bar\zeta_{\dot\beta}}{\langle \iota\zeta \rangle^2} \\ &\quad \times \int_0^\infty \omega d\omega \left(e^{-i\omega x^{\alpha\dot\alpha}\zeta_\alpha\bar\zeta_{\dot\alpha}} a_-(\omega, \zeta, \bar\zeta) + e^{i\omega x^{\alpha\dot\alpha}\zeta_\alpha\bar\zeta_{\dot\alpha}} a_+^\dagger(\omega, \zeta, \bar\zeta)\right) \\ &= \frac{i\kappa}{16\pi^3} \int_0^\infty \omega d\omega \\ &\quad \times \int_{\mathbb{CP}^1} d\zeta d\bar\zeta \, \epsilon^{(+)}_{\alpha\dot\alpha\beta\dot\beta}(\zeta, \bar\zeta) \left(e^{-i\omega x^{\alpha\dot\alpha}\zeta_\alpha\bar\zeta_{\dot\alpha}} a_-(\omega, \zeta, \bar\zeta) + e^{i\omega x^{\alpha\dot\alpha}\zeta_\alpha\bar\zeta_{\dot\alpha}} a_+^\dagger(\omega, \zeta, \bar\zeta)\right). \end{aligned} \tag{25}$$

Here, in order not to confuse it with the spacetime coordinate $\lambda_\alpha = (1, z)$, the integration variable has been taken to be $\zeta_\alpha = (1, \zeta)$.

The negative helicity linearized Weyl tensor is recovered from the twistor representative $\widetilde{\mathbf{h}} \in \Omega^{0,1}(\mathbb{PT}, \mathcal{O}(-6))$, given by (20), as:

$$\begin{aligned} \overline{\Psi}_{\alpha\beta\gamma\delta}(x) &= \frac{i}{2\pi} \int_{\mathbb{CP}^1} \langle \zeta d\zeta \rangle \, \zeta_\alpha \zeta_\beta \zeta_\gamma \zeta_\delta \, \widetilde{\mathbf{h}}(\mu^{\dot\alpha} = x^{\alpha\dot\alpha}\zeta_\alpha, \zeta_\alpha) \\ &= \frac{i\kappa}{16\pi^3} \int_0^\infty \omega^3 d\omega \\ &\quad \times \int_{\mathbb{CP}^1} d\zeta d\bar\zeta \, \zeta_\alpha \zeta_\beta \zeta_\gamma \zeta_\delta \left(e^{-i\omega x^{\alpha\dot\alpha}\zeta_\alpha\bar\zeta_{\dot\alpha}} a_+(\omega, \zeta, \bar\zeta) + e^{i\omega x^{\alpha\dot\alpha}\zeta_\alpha\bar\zeta_{\dot\alpha}} a_-^\dagger(\omega, \zeta, \bar\zeta)\right). \end{aligned} \tag{26}$$

---

[5]The Weyl tensor would have been obtained as $\Psi_{\dot\alpha\dot\beta\dot\gamma\dot\delta}(x) = \frac{i}{2\pi} \int_{\mathbb{CP}^1} \langle \zeta d\zeta \rangle \wedge \frac{\partial^4 \mathbf{h}}{\partial\mu^{\dot\alpha}\partial\mu^{\dot\beta}\partial\mu^{\dot\gamma}\partial\mu^{\dot\delta}}(\mu^{\dot\alpha} = x^{\alpha\dot\alpha}\zeta_\alpha, \zeta_\alpha)$. Note the sign difference which ultimately mirror the fact that $\Psi_4^0 = -\partial_u^2 \bar\sigma$.

***iii*) Large $r$ limit: $\mathbb{M} \to \mathscr{I}$**

Contracting (25) with $(\partial_{\bar{z}})^{\alpha\dot{\alpha}}$ and making use of (9), we obtain

$$
\begin{aligned}
\Phi_{\bar{z}\bar{z}}(x) &= \frac{r}{2\pi i}\int_{\mathbb{CP}^1}\langle\zeta d\zeta\rangle\wedge\frac{\langle\iota\lambda\rangle^2}{\langle\iota\zeta\rangle^2}n^\alpha n^\beta\frac{\partial^2\mathbf{h}}{\partial\mu^{\dot{\alpha}}\partial\mu^{\dot{\beta}}}(\mu^{\dot{\alpha}}=x^{\alpha\dot{\alpha}}\zeta_\alpha,\zeta_\alpha) \\
&= \frac{\kappa r^2}{8\pi^3}\int_0^\infty\omega d\omega\int_{\mathbb{CP}^1}\frac{i}{2}d\zeta d\bar{\zeta}\,\frac{\langle\iota\lambda\rangle^2}{\langle\iota\zeta\rangle^2}\left(e^{-i\omega x^{\alpha\dot{\alpha}}\zeta_\alpha\bar{\zeta}_{\dot{\alpha}}}a_-(\omega,\zeta,\bar{\zeta})+e^{i\omega x^{\alpha\dot{\alpha}}\zeta_\alpha\bar{\zeta}_{\dot{\alpha}}}a_+^\dagger(\omega,\zeta,\bar{\zeta})\right).
\end{aligned}
\tag{27}
$$

Taking the large $r$ limit and making use of the saddle point approximation or the identity $e^{\mp i\omega x^{\alpha\dot{\alpha}}\zeta_\alpha\bar{\zeta}_{\dot{\alpha}}}=\mp\frac{\pi i}{\omega r}e^{\mp i\omega u}\delta(z-\zeta)+\mathcal{O}(r^{-2})$, one recovers[6] the asymptotic shear (12) from which we started, namely

$$
\begin{aligned}
\bar{\sigma}(u,\lambda,\bar{\lambda}) &= \lim_{r\to\infty}r^{-1}\Phi_{\bar{z}\bar{z}}(x) \\
&= -\frac{i\kappa}{8\pi^2}\int_0^\infty d\omega\left(e^{-i\omega u}a_-(\omega,\lambda,\bar{\lambda})-e^{i\omega u}a_+^\dagger(\omega,\lambda,\bar{\lambda})\right),
\end{aligned}
\tag{28}
$$

as expected.

Similarly, for the opposite helicity, we contract (26) with $n^\alpha$ to obtain $\overline{\Psi}_4$

$$
\begin{aligned}
\overline{\Psi}_4(x) &= n^\alpha n^\beta n^\gamma n^\delta\overline{\Psi}_{\alpha\beta\gamma\delta}(x) \\
&= \frac{i}{2\pi}\int_{\mathbb{CP}^1}\langle\zeta d\zeta\rangle\wedge\widetilde{\mathbf{h}}(\mu^{\dot{\alpha}}=x^{\alpha\dot{\alpha}}\zeta_\alpha,\zeta_\alpha) \\
&= \frac{\kappa}{8\pi^3}\int_0^\infty\omega^3 d\omega\int_{\mathbb{CP}^1}\frac{i}{2}d\zeta d\bar{\zeta}\left(e^{-i\omega x^{\alpha\dot{\alpha}}\zeta_\alpha\bar{\zeta}_{\dot{\alpha}}}a_+(\omega,\zeta,\bar{\zeta})+e^{i\omega x^{\alpha\dot{\alpha}}\zeta_\alpha\bar{\zeta}_{\dot{\alpha}}}a_-^\dagger(\omega,\zeta,\bar{\zeta})\right),
\end{aligned}
\tag{29}
$$

and recover $\sigma(u,\lambda,\bar{\lambda})$ as given in (13) from [64,65]

$$
\begin{aligned}
\partial_u^2\sigma(u,\lambda,\bar{\lambda}) &= -\overline{\Psi}_4^0(u,\lambda,\bar{\lambda}) \\
&= -\lim_{r\to\infty}r\overline{\Psi}_4(x) \\
&= \frac{i\kappa}{8\pi^2}\int_0^\infty\omega^2 d\omega\left(e^{-i\omega u}a_+(\omega,\zeta,\bar{\zeta})-e^{i\omega u}a_-^\dagger(\omega,\zeta,\bar{\zeta})\right),
\end{aligned}
\tag{30}
$$

as it should.

## 3 Carrollian representation of the $\mathscr{L}w_{1+\infty}$ algebra

### 3.1 Representation of the $\mathscr{L}w_{1+\infty}$ algebra

We now turn to the action of $\mathscr{L}w_{1+\infty}$ algebra on Carrollian fields living at $\mathscr{I}$.

**The $\mathscr{L}w_{1+\infty}$ algebra**

The generators $g(Z^A,\bar{Z}^A)$ of the $\mathscr{L}w_{1+\infty}$ algebra, as realized in twistor space, are the functions on $\mathbb{T}$ of homogeneity degree 2 (in $Z^A=(\mu^{\dot{\alpha}},\lambda_\alpha)$) such that

$$
g = \underbrace{g_0(z)}_{n=0}+\underbrace{g_{\dot{\alpha}}(z)\mu^{\dot{\alpha}}}_{n=1}+\underbrace{g_{\dot{\alpha}(2)}(z)\mu^{\dot{\alpha}(2)}}_{n=2}+\dots,
\tag{31}
$$

---

[6]Note the $\delta$-function normalisation $\int\frac{i}{2}d\zeta d\bar{\zeta}\,\delta(\zeta)=1$.

with

$$g_{\dot{\alpha}(n)}(z) = \sum_{k=-\infty}^{+\infty} g_{\dot{\alpha}(n)}^{(k)} z^k, \tag{32}$$

some holomorphic functions on $\mathbb{C}^* = \mathbb{C}\backslash\{0\}$. The generators are decomposed into polynomials $g_{\dot{\alpha}(n)}(z)\mu^{\dot{\alpha}(n)}$ of degree $n \in \mathbb{N}$ on the plane of coordinates $\mu^{\dot{\alpha}} = (\mu^{\dot{0}}, \mu^{\dot{1}})$. Here and everywhere in this article we make use of the notation (commonly used in the higher-spin literature) $\alpha(n) := (\alpha_1 \alpha_2 \ldots \alpha_n)$ for $n$ symmetrized indices and similarly $\mu^{\dot{\alpha}(n)} = \mu^{\dot{\alpha}_1} \ldots \mu^{\dot{\alpha}_n}$.

The $\mathscr{L}w_{1+\infty}$ algebra is explicitly realized through the holomorphic Poisson bracket $\epsilon = \epsilon^{\dot{\alpha}\dot{\beta}} \frac{\partial}{\partial \mu^{\dot{\alpha}}} \frac{\partial}{\partial \mu^{\dot{\beta}}}$ [27]

$$\{g_1, g_2\} := \epsilon^{\dot{\alpha}\dot{\beta}} \frac{\partial g_1}{\partial \mu^{\dot{\alpha}}} \frac{\partial g_2}{\partial \mu^{\dot{\beta}}}. \tag{33}$$

It can be alternatively expressed in terms of the modes

$$w_m^p := (\mu^{\dot{0}})^{p+m-1} (\mu^{\dot{1}})^{p-m-1}, \qquad |m| \le p-1, \tag{34}$$

with $p = \frac{n+2}{2}$ as (see e.g. [22] in the celestial literature)

$$\{w_m^p, w_n^q\} = 2(m(q-1) - n(p-1)) w_{m+n}^{p+q-2}. \tag{35}$$

**Carrollian fields**

A Carrollian field $\phi(u, z, \bar{z}) \in \mathcal{C}_{(k,\bar{k})}^{\infty}(\mathscr{I})$ of weight $(k, \bar{k}) = (\frac{1-s}{2}, \frac{1+s}{2})$ lifts, through the map (22), to an holomorphic form $\mathbf{f} = f \bar{D}\bar{\lambda} \in \Omega^{0,1}(\mathbb{PT}, \mathcal{O}(2s-2))$ in twistor space. The action of the $\mathscr{L}w_{1+\infty}$ algebra is then given by (see [27][7])

$$\delta_g \mathbf{f} = \{g, \mathbf{f}\} = \{g, f\} \bar{D}\bar{\lambda}. \tag{36}$$

Since $g$ is generically singular at $z = 0$ and $z = \infty$, the resulting twistor field $\delta_g \mathbf{f}(\mu^{\dot{\alpha}}, \lambda_\alpha)$ will be singular. This is a problem as it might render the Penrose transform ill-defined or generate a singularity in the resulting field. It will thus be useful to restrict ourselves to Carrollian fields with support inside an annulus

$$\mathcal{A} = \left\{ z \in \mathbb{C} \quad \text{s.t.} \quad \frac{1}{R} < |z| < R \right\},$$

where $R$ is some fixed number that can be taken as large as we want. We will denote such fields $\phi(u, z, \bar{z}) \in \mathcal{C}_{(k,\bar{k})}^{\infty}(\mathscr{I}_{\mathcal{A}})$. These are such that, for example, (36) is regular on the whole of $S^2$. A perhaps more physical justification for this space of fields is the following: as was pointed out in [27], due to Dolbeault-Cech equivalence of cohomology, the generator (31) of a symmetry can be equivalently realized as the twistor representative $\bar{d}g$ of a linearized field; the presence of a singularity now being essential to ensure that the corresponding cohomology class is non trivial. Accordingly, (36) can be thought of as the action of a graviton on an other and the above restriction amounts to requiring that these are not inserted at the same point of the celestial sphere (thus avoiding a type of collinear singularity).

---

[7]In this reference the symmetry is thought of as a deformation of the complex structure, which yields an extra inhomogeneous term $\bar{d}g$. In this article, we restrict ourselves to the linear action on the linearized fields. This action will induce a representation on the Carrollian fields and, by construction, this excludes an inhomogeneous (soft) shift.

**Carrollian representation of the algebra**

The action $\delta_g \phi(u, z, \bar{z})$ of a generator (31) on a Carrollian field $\phi(u, z, \bar{z}) \in \mathcal{C}^\infty_{(k,\bar{k})}(\mathscr{I}_{\mathcal{A}})$ of weight $(k, \bar{k}) = (\frac{1-s}{2}, \frac{1+s}{2})$, with s = ±2, is obtained by following the successive steps of Fig. 1: i) First the Carrollian field $\phi(u, z, \bar{z})$ is lifted to the preferred twistor representative $\mathbf{f} := \mathbf{T}^s(\phi) \in \Omega^{0,1}(\mathbb{PT}, \mathcal{O}(2s - 2))$ via the map (22). The generator g of the algebra then acts on this representative as (36). ii) Second the Penrose transform of $\delta_g \mathbf{f}$ yields a solution $\delta_g \Phi(x)$ of the zero-rest-mass equation of helicity s. iii) Third, taking the limit $r \to \infty$ we obtain the corresponding Carrollian field $\delta_g \phi(u, z, \bar{z})$.

In section 5 we go through this procedure for a Carrollian field $\bar{\sigma} \in \mathcal{C}^\infty_{(-\frac{1}{2}, \frac{3}{2})}(\mathscr{I}_{\mathcal{A}})$ corresponding to a field of helicity +2 (respectively for $\sigma \in \mathcal{C}^\infty_{(\frac{3}{2}, -\frac{1}{2})}(\mathscr{I}_{\mathcal{A}})$, corresponding to a field of helicity −2) and derive the following.

> **Proposition 1.** *The action of the generator $g_{\dot{\alpha}(n)}(z)$ of the $\mathscr{L}w_{1+\infty}$ algebra on the Carrollian fields of spin 2, $\bar{\sigma}(u, z, \bar{z})$ and $\sigma(u, z, \bar{z})$, is given by the following:*
>
> $$\delta_n \bar{\sigma} = \sum_{\ell=0}^{n} \bar{\partial}^{n-\ell} \Big( g_{\dot{\alpha}(n)} \bar{\lambda}^{\dot{\alpha}(n)} \Big) \frac{\ell}{(n-\ell)!} \partial_u^3 \Big( u^{n-\ell} \partial_u^{-1-\ell} \bar{\partial}^{\ell-1} \bar{\sigma} \Big),$$
> $$\delta_n \sigma = \sum_{\ell=0}^{n} \bar{\partial}^{n-\ell} \Big( g_{\dot{\alpha}(n)} \bar{\lambda}^{\dot{\alpha}(n)} \Big) \frac{\ell}{(n-\ell)!} \partial_u^{-1} \Big( u^{n-\ell} \partial_u^{3-\ell} \bar{\partial}^{\ell-1} \sigma \Big),$$
>
> (37)
>
> *where $n \in \mathbb{N}$ and $\bar{\partial} := \frac{\partial}{\partial \bar{z}}$.*

*Proof.* See section 5. □

The proposition can also be rephrased as follows: for a Carrollian field $\phi$ of weight $(\frac{1+s}{2}, \frac{1-s}{2})$, corresponding to a zero-rest-mass field of helicity s = ±2, we can write the action of the generators as

$$\delta_n \phi = \sum_{\ell=0}^{n} \bar{\partial}^{n-\ell} \Big( g_{\dot{\alpha}(n)} \bar{\lambda}^{\dot{\alpha}(n)} \Big) \frac{\ell}{(n-\ell)!} \partial_u^{1+s} \Big( u^{n-\ell} \partial_u^{1-s-\ell} \bar{\partial}^{\ell-1} \phi \Big), \tag{38}$$

and it is tempting to conjecture that this formula extends to any spin $s \in \mathbb{Z}$. Even though our method of derivation in principle applies to any spin in practice the computation is rather lengthy and in this article we will only explicitly show the proof for the spin-two case formula. Nevertheless, as it should be clear from the proof of the proposition below, the representation itself does extend to any spin. Let us emphasize, though, that the corresponding representation on spin-one fields should not be confused with the infinitesimal action of the symmetries of self-dual Yang-Mills in twistor space (as e.g. in [6,7]) nor with the one appearing in the celestial gluon OPE [22–24]. Rather, these are the extension of the gravitational twistor symmetry to other spins (this is similar to the fact that the BMS group acts on all fields regardless of their helicity but e.g. is not the group of asymptotic symmetries of QED).

The action (38) is obviously linear (and as such does not contain an inhomogeneous (soft) shift). In fact, as we shall now see, it forms a representation of the algebra.

**Proposition 2.** *The action of the generators (31) on Carrollian fields $\mathcal{C}^\infty_{(k,\bar{k})}(\mathscr{I}_{\mathcal{A}})$ of weight $(k, \bar{k}) = (\frac{1-s}{2}, \frac{1+s}{2})$, as defined through the procedure explained before Proposition 1, forms a representation of the $\mathscr{L}w_{1+\infty}$ algebra. In particular the action (37) is a representation of $\mathscr{L}w_{1+\infty}$.*

*Proof.*
Let $\phi(u,z,\bar{z})$ be a Carrollian field of weight $(k,\bar{k}) = (\frac{1-s}{2}, \frac{1+s}{2})$. Via the map (22) it defines a preferred twistor representative $\mathbf{f} = \mathbf{T}^s(\phi) \in \Omega^{0,1}(\mathbb{PT}, \mathcal{O}(2s-2))$. In this proof only, let us introduce a shorthand notation that will be very useful: if $\mathbf{A} \in \Omega^{0,1}(\mathbb{PT}, \mathcal{O}(2s-2))$ is any twistor representative we will denote by $\widetilde{\mathbf{A}}(u,z,\bar{z})$ the Carrollian field of weight $(k,\bar{k}) = (\frac{1-s}{2}, \frac{1+s}{2})$ obtained from $\mathbf{A}$ by successive Penrose transform and large $r$ expansion. For example we have, by construction, $\widetilde{\mathbf{f}}(u,z,\bar{z}) = \phi(u,z,\bar{z})$. Let $\delta_{g_i}\mathbf{f} = \{f, g_i\}\bar{D}\bar{\lambda}$ be the twistor representative resulting from the action of elements $g_1$, $g_2$, on $\mathbf{f}$ and let $\delta_{g_i}\phi(u,z,\bar{z}) := \widetilde{\delta_{g_i}\mathbf{f}}(u,z,\bar{z})$ be the corresponding action on the Carrollian field. We will note $\mathbf{f}_i = \mathbf{T}^s(\delta_{g_i}\phi)$ the preferred twistor representative associated to $\delta_{g_i}\phi(u,z,\bar{z})$. Both $\delta_{g_i}\mathbf{f}$ and $\mathbf{f}_i$ define the same spacetime fields through the Penrose transform,

$$\delta_{g_i}\phi(u,z,\bar{z}) = \widetilde{\delta_{g_i}\mathbf{f}}(u,z,\bar{z}) = \widetilde{\mathbf{f}_i}(u,z,\bar{z}),$$

but they do not have to coincide in general. Rather these two representatives must be in the same cohomology class (this is because of the isomorphism (24) underlying the Penrose transform, see [59] for a proof). This means that

$$\delta_{g_i}\mathbf{f} = \mathbf{f}_i + \bar{\mathrm{d}}\alpha_i,$$

where $\alpha_1$, $\alpha_2$ are some functions on twistor space. Importantly, since both $\delta_{g_i}\mathbf{f}$ and $\mathbf{f}_i$ only have support on the annulus $\mathcal{A}$ so do $\alpha_1$ and $\alpha_2$. Now by definition $\delta_{g_2}\delta_{g_1}\phi(u,z,\bar{z}) = \widetilde{\delta_{g_2}\mathbf{f}_1}(u,z,\bar{z})$ and thus

$$\delta_{g_2}\delta_{g_1}\phi(u,z,\bar{z}) = \widetilde{\delta_{g_2}\delta_{g_1}\mathbf{f}}(u,z,\bar{z}) - \widetilde{\delta_{g_2}\bar{\mathrm{d}}\alpha_1}(u,z,\bar{z}).$$

In order to prove that we have a representation of the algebra, we need to prove that the Carrollian field $\delta_{\{g_1,g_2\}}\phi(u,z,\bar{z}) = \widetilde{\delta_{\{g_1,g_2\}}\mathbf{f}}(u,z,\bar{z})$ obtained from $\delta_{\{g_1,g_2\}}\mathbf{f} = (\delta_{g_1}\delta_{g_2} - \delta_{g_2}\delta_{g_1})\mathbf{f}$ coincides with $(\delta_{g_1}\delta_{g_2} - \delta_{g_2}\delta_{g_1})\phi(u,z,\bar{z})$. We therefore need to prove that,

$$
\begin{aligned}
&\delta_{\{g_1,g_2\}}\phi(u,z,\bar{z}) - (\delta_{g_1}\delta_{g_2} - \delta_{g_2}\delta_{g_1})\phi(u,z,\bar{z}) \\
&= \widetilde{\delta_{\{g_1,g_2\}}\mathbf{f}}(u,z,\bar{z}) - (\delta_{g_1}\delta_{g_2} - \delta_{g_2}\delta_{g_1})\phi(u,z,\bar{z}) \\
&= \left(\widetilde{\delta_{g_1}\delta_{g_2}\mathbf{f}}(u,z,\bar{z}) - \widetilde{\delta_{g_2}\delta_{g_1}\mathbf{f}}(u,z,\bar{z})\right) - \left(\delta_{g_1}\delta_{g_2}\phi(u,z,\bar{z}) - \delta_{g_2}\delta_{g_1}\phi(u,z,\bar{z})\right) \\
&= \widetilde{\delta_{g_1}\bar{\mathrm{d}}\alpha_2}(u,z,\bar{z}) + \widetilde{\delta_{g_2}\bar{\mathrm{d}}\alpha_1}(u,z,\bar{z}),
\end{aligned}
$$

vanishes. We will in fact prove that the terms $\delta_g\bar{\mathrm{d}}\alpha$ are $\bar{\mathrm{d}}$-exact, since the Penrose transform annihilates exact forms this will be enough to conclude. To see this, we note that

$$\delta_g\bar{\mathrm{d}}\alpha = \{g, \bar{\mathrm{d}}\alpha\} = \bar{\mathrm{d}}\{g, \alpha\} - \{\bar{\mathrm{d}}g, \alpha\}.$$

Now $\bar{\mathrm{d}}g$ has only support on $z = 0$ and $z = \infty$, while $\alpha$ only has support on the annulus $\mathcal{A}$ therefore the last term vanishes. Finally since $g$ is non singular on $\mathcal{A}$ and $\alpha$ only has support on this set it follows that $\{g, \alpha\}$ is a non singular function on the whole of $S^2$. Thus $\delta_g\bar{\mathrm{d}}\alpha = \bar{\mathrm{d}}\{g, \alpha\}$ is $\bar{\mathrm{d}}$-exact and $\widetilde{\delta_g\bar{\mathrm{d}}\alpha}(u,z,\bar{z}) = 0$.

$\square$

Note that, even though the action (36) would be admissible for any $g(z,\bar{z})$, holomorphicity on $\mathcal{A}$ of $g$, $\bar{\partial}g = 0$, is crucial in the above proof.

## 3.2 Action of the simplest generators $n = 1, 2, 3$

In order to illustrate the general expression of Proposition 1, let us write down the action of the simplest generators.[8]

---

[8]Notice that the $n = 0$ ($p = 1$) generator acts trivially.

**$n = 1$**

$$\delta_1 \bar{\sigma} = \left[ g_{\dot{\alpha}_1} \bar{\lambda}^{\dot{\alpha}_1} \partial_u \right] \bar{\sigma}, \qquad \delta_1 \sigma = \left[ g_{\dot{\alpha}_1} \bar{\lambda}^{\dot{\alpha}_1} \partial_u \right] \sigma. \tag{39}$$

This coincides with the usual action of a supertranslation vector field which can be expanded in modes as

$$\tau_{g_1}(z, \bar{z}) \partial_u = g_{\dot{\alpha}_1} \bar{\lambda}^{\dot{\alpha}_1} \partial_u = \left( \sum_{m=-\infty}^{\infty} \sum_{\bar{m}=0}^{1} T_{m,\bar{m}} z^m \bar{z}^{\bar{m}} \right) \partial_u. \tag{40}$$

**$n = 2$**

From

$$\delta_2 \bar{\sigma} = \left[ \bar{\partial} \left( 2 g_{\dot{\alpha}(2)} \bar{\lambda}^{\dot{\alpha}(2)} \right) \left( \frac{3}{2} + \frac{u}{2} \partial_u \right) + \left( 2 g_{\dot{\alpha}(2)} \bar{\lambda}^{\dot{\alpha}(2)} \right) \bar{\partial} \right] \bar{\sigma},$$
$$\delta_2 \sigma = \left[ \bar{\partial} \left( 2 g_{\dot{\alpha}(2)} \bar{\lambda}^{\dot{\alpha}(2)} \right) \left( -\frac{1}{2} + \frac{u}{2} \partial_u \right) + \left( 2 g_{\dot{\alpha}(2)} \lambda^{\dot{\alpha}(2)} \right) \bar{\partial} \right] \sigma, \tag{41}$$

one recognizes the usual action of the vector field,

$$\tau_{g_2}(z, \bar{z}) \bar{\partial} = 2 g_{\dot{\alpha}\dot{\beta}}(z) \bar{\lambda}^{\dot{\alpha}} \bar{\lambda}^{\dot{\beta}} \bar{\partial} = \left( \sum_{m=-\infty}^{\infty} \sum_{\bar{m}=0}^{2} L_{m,\bar{m}} z^m \bar{z}^{\bar{m}} \right) \bar{\partial}, \tag{42}$$

on the shear. To interpret this, it is here useful to read from (33) with $n = 2$ the algebra,

$$\left\{ g_{1\alpha\beta}(z, \bar{z}) \mu^\alpha \mu^\beta, \, g_{2\alpha\beta}(z, \bar{z}) \mu^\alpha \mu^\beta \right\} = 4 g_{1\gamma\alpha}(z, \bar{z}) g_2{}^\gamma{}_\beta(z, \bar{z}) \mu^\alpha \mu^\beta. \tag{43}$$

When the generators are globally holomorphic, this is the SL(2, ℂ) algebra. Under the more general assumption (32) that the generators admit Laurent series on ℂ*, this is the SL(2, ℂ)-loop algebra [66]. A direct computation shows that it is isomorphic to the algebra of vector fields on ℂ* of the form (42).

**$n = 3$**

Let us finally illustrate the action for the (less familiar) $n = 3$ generator,

$$\delta_3 \bar{\sigma} = \left[ \left( \bar{\lambda}^{\dot{\alpha}(3)} g_{\dot{\alpha}(3)} \right) 3 \partial_u^{-1} \bar{\partial}^2 + \bar{\partial} \left( g_{\dot{\alpha}(3)} \bar{\lambda}^{\dot{\alpha}(3)} \right) \left( 2u + 6 \partial_u^{-1} \right) \bar{\partial} + \bar{\partial}^2 \left( g_{\dot{\alpha}(3)} \bar{\lambda}^{\dot{\alpha}(3)} \right) \left( 3u + 3 \partial_u^{-1} + \frac{1}{2} u^2 \partial_u \right) \right] \bar{\sigma},$$
$$\delta_3 \sigma = \left[ \left( \bar{\lambda}^{\dot{\alpha}(3)} g_{\dot{\alpha}(3)} \right) 3 \partial_u^{-1} \bar{\partial}^2 + \bar{\partial} \left( g_{\dot{\alpha}(3)} \bar{\lambda}^{\dot{\alpha}(3)} \right) \left( 2u - 2 \partial_u^{-1} \right) \bar{\partial} + \bar{\partial}^2 \left( g_{\dot{\alpha}(3)} \bar{\lambda}^{\dot{\alpha}(3)} \right) \left( -u + \partial_u^{-1} + \frac{1}{2} u^2 \partial_u \right) \right] \sigma. \tag{44}$$

This action corresponds to the one of the sub-subleading current labelled by $s = 2$ in [47, 48] and denoted by $p = \frac{5}{2}$ in [22].

## 4 Canonical action of charges

In this section, we show that the action of the $\mathscr{L}w_{1+\infty}$ symmetries at null infinity derived in Proposition 1 coincides with the action of the canonical charges on the radiative data that was derived from gravitational phase space methods in [48]. The dictionary is that the label $s$ referred to as the spin there is related to the generator index as $s = n - 1 = 2p - 3$ (hence supertranslations are spin 0, superrotations spin 1, etc.).

## 4.1 Canonical charges

Let us first review the prescription of [48] for the construction of spin-$s$ charges $\mathcal{Q}_s$.[9] As part of the procedure, a first set $\mathcal{Q}_s$ for $s \geq -2$ is defined by solving the following recursion relation

$$\mathcal{Q}_s(u,z,\bar{z}) = \left(\partial_u^{-1}\bar{\partial}\right)\mathcal{Q}_{s-1} + \frac{s+1}{2}\partial_u^{-1}\left[\bar{\sigma}\mathcal{Q}_{s-2}\right], \tag{45}$$

starting from $\mathcal{Q}_{-2} := \frac{1}{2}\partial_u^2\sigma$. The hard charges, denoted by $\mathcal{Q}_s^2$, are then obtained by only keeping the terms in $\mathcal{Q}_s$ which are quadratic (hence the superscript 2) in the shear $\bar{\sigma}, \sigma$. The result is

$$\mathcal{Q}_s^2(u,z,\bar{z}) = \frac{1}{4}\sum_{\ell=0}^{s}(\ell+1)\partial_u^{-1}\left(\partial_u^{-1}\bar{\partial}\right)^{s-\ell}\left[\bar{\sigma}\left(\partial_u^{-1}\bar{\partial}\right)^{\ell}\partial_u^2\sigma\right]. \tag{46}$$

It receives the following interpretation: for $s = -2,-1,0,1,2$ each of the quantities $\mathcal{Q}_s$ is proportional to the Newman-Penrose scalars $\Psi_4^0, \Psi_3^0, \Psi_2^0, \Psi_1^0, \Psi_0^0$ and the recursion relations (45) correspond to their associated Bianchi identities [65]. The charges $\mathcal{Q}_s^2$ for generic $s$ are thus quadratic quantities in the shear that generalize the behavior of the (quadratic part of) Newman-Penrose scalars.

The action of these charges on the shear would generically be divergent (see [47]); for this reason one introduces the following renormalization prescription

$$\hat{q}_s^2(u,z,\bar{z}) := \sum_{n=0}^{s}\frac{(-u)^{s-n}}{(s-n)!}\bar{\partial}^{s-n}\mathcal{Q}_n^2(u,z,\bar{z}), \tag{47}$$

which will define higher-spin charge aspects $q_s^2(z,\bar{z}) = \lim_{u\to-\infty}\hat{q}_s^2(u,z,\bar{z})$. Using (46), one then has

$$\hat{q}_s^2(u,z,\bar{z}) = \frac{1}{4}\sum_{n=0}^{s}\sum_{\ell=0}^{n}\frac{(\ell+1)(-u)^{s-n}}{(s-n)!}\partial_u^{-(n-\ell+1)}\bar{\partial}^{s-\ell}\left[\bar{\sigma}\left(\partial_u^{-1}\bar{\partial}\right)^{\ell}\partial_u^2\sigma\right]. \tag{48}$$

Ashtekar-Streubel's symplectic structure [69]

$$\{\partial_u\sigma(u,z,\bar{z}),\bar{\sigma}(u',z',\bar{z}')\} = \frac{\kappa^2}{2}\delta(u-u')\delta(z-z'), \tag{49}$$

then allows to compute the action of the charges as

$$\begin{aligned}\{q_s^2(z,\bar{z}),\bar{\sigma}(u',z',\bar{z}')\} &= \lim_{u\to-\infty}\{\hat{q}_s^2(u,z,\bar{z}),\bar{\sigma}(u',z',\bar{z}')\}, \\ \{q_s^2(z,\bar{z}),\sigma(u',z',\bar{z}')\} &= \lim_{u\to-\infty}\{\hat{q}_s^2(u,z,\bar{z}),\sigma(u',z',\bar{z}')\}.\end{aligned} \tag{50}$$

The result is[10] [48]

$$\begin{aligned}\{q_s^2(z,\bar{z}),\bar{\sigma}(u',z',\bar{z}')\} &= \frac{\kappa^2}{8}\sum_{n=0}^{s}(-1)^{s+n}\frac{(n+1)(\hat{\Delta}+2)_{s-n}}{(s-n)!}\partial_{u'}^{1-s}\bar{\partial}^n\bar{\sigma}(u',z',\bar{z}')\bar{\partial}^{s-n}\delta(z-z'), \\ \{q_s^2(z,\bar{z}),\sigma(u',z',\bar{z}')\} &= \frac{\kappa^2}{8}\sum_{n=0}^{s}(-1)^{s+n}\frac{(n+1)(\hat{\Delta}-2)_{s-n}}{(s-n)!}\partial_{u'}^{1-s}\bar{\partial}^n\sigma(u',z',\bar{z}')\bar{\partial}^{s-n}\delta(z-z'),\end{aligned} \tag{51}$$

where $\hat{\Delta} := u\partial_u + 1$ and $(x)_n = x(x-1)\cdots(x-n+1)$ is the falling factorial.

---

[9]The dictionary to convert the expressions of this reference into ours is $C \mapsto \bar{\sigma}$ and $D := \frac{1}{\sqrt{2}}\eth \mapsto \bar{\partial}$. Note again that, as compared to the usual definition of the shear, we have $\bar{\sigma}^{\text{here}} = C_{\bar{z}\bar{z}} = 2\bar{\sigma}^{\text{NP}}$; see [67,68] for a remainder of Newman-Penrose (NP) conventions.

[10]See eq. (67) and (68).

## 4.2 Matching with the twistor space action

The integration of (51) against a function $\tau_s(z,\bar{z})$ on the sphere yields the actions of a generator

$$
\begin{aligned}
\delta_{\tau_s}\bar{\sigma}(u,z,\bar{z}) &= \left\{ \frac{8}{\kappa^2} \int_{S^2} d\zeta^2 \tau_s(\zeta,\bar{\zeta}) q_s^2(\zeta,\bar{\zeta}),\ \bar{\sigma}(u,z,\bar{z}) \right\}, \\
\delta_{\tau_s}\sigma(u,z,\bar{z}) &= \left\{ \frac{8}{\kappa^2} \int_{S^2} d\zeta^2 \tau_s(\zeta,\bar{\zeta}) q_s^2(\zeta,\bar{\zeta}),\ \sigma(u,z,\bar{z}) \right\},
\end{aligned}
\tag{52}
$$

which reads

$$
\begin{aligned}
\delta_{\tau_s}\bar{\sigma}(u,z) &= \sum_{\ell=0}^{s} \frac{(\ell+1)(\hat{\Delta}+2)_{s-\ell}}{(s-\ell)!} (\bar{\partial}^{s-\ell}\tau_s)\bar{\partial}^{\ell}\partial_u^{1-s}\bar{\sigma}(u,z), \\
\delta_{\tau_s}\sigma(u,z) &= \sum_{\ell=0}^{s} \frac{(\ell+1)(\hat{\Delta}-2)_{s-\ell}}{(s-\ell)!} (\bar{\partial}^{s-\ell}\tau_s)\bar{\partial}^{\ell}\partial_u^{1-s}\sigma(u,z).
\end{aligned}
\tag{53}
$$

In order to relate these expressions with the results of section 3, it is useful to note the following identities [48]

$$
\begin{aligned}
u^n \partial_u^n &= (\hat{\Delta}-1)_n, & \partial_u^n u^n &= (\hat{\Delta}+n-1)_n, \\
u^{-n}\partial_u^{-n} &= (\hat{\Delta}+n-1)_n^{-1}, & & \\
\partial_u(\hat{\Delta}+\alpha)_n &= (\hat{\Delta}+\alpha+1)_n\partial_u, & \partial_u^{-1}(\hat{\Delta}+\alpha)_n &= (\hat{\Delta}+\alpha-1)_n\partial_u^{-1}, \\
u(\hat{\Delta}+\alpha)_n &= (\hat{\Delta}+\alpha-1)_n u, & u(\hat{\Delta}+n-1)_n^{-1} &= (\hat{\Delta}+n-2)_n^{-1}u.
\end{aligned}
\tag{54}
$$

They lead to

$$
\frac{(\hat{\Delta}+2)_{s-\ell}}{(s-\ell)!} = \partial_u^3 \left( \frac{(\hat{\Delta}-1)_{s-\ell}}{(s-\ell)!} \right) \partial_u^{-3} = \partial_u^3 \left( \frac{u^{s-\ell}}{(s-\ell)!}\partial_u^{s-\ell} \right) \partial_u^{-3},
\tag{55}
$$

and, similarly, one has

$$
\frac{(\hat{\Delta}-2)_{s-\ell}}{(s-\ell)!} = \partial_u^{-1}\frac{(\hat{\Delta}-1)_{s-\ell}}{(s-\ell)!}\partial_u = \partial_u^{-1}\left( \frac{u^{s-\ell}\partial_u^{s-\ell}}{(s-\ell)!} \right)\partial_u.
\tag{56}
$$

We can therefore rewrite the action (53) as

$$
\begin{aligned}
\delta_{\tau_s}\bar{\sigma}(u,z) &= \sum_{\ell=0}^{s} \frac{(\ell+1)}{(s-\ell)!} (\bar{\partial}^{s-\ell}\tau_s)\partial_u^3\left( u^{s-\ell}\partial_u^{-2-\ell}\bar{\partial}^{\ell}\bar{\sigma}(u,z) \right), \\
\delta_{\tau_s}\sigma(u,z) &= \sum_{\ell=0}^{s} \frac{(\ell+1)}{(s-\ell)!} (\bar{\partial}^{s-\ell}\tau_s)\partial_u^{-1}\left( u^{s-\ell}\partial_u^{2-\ell}\bar{\partial}^{\ell}\sigma(u,z) \right).
\end{aligned}
\tag{57}
$$

Using the relationship $s = n-1$, they read

$$
\begin{aligned}
\delta_{\tau_{n-1}}\bar{\sigma}(u,z) &= \sum_{\ell=1}^{n} \frac{\ell}{(n-\ell)!} (\bar{\partial}^{n-\ell}\tau_{n-1})\partial_u^3\left( u^{n-\ell}\partial_u^{-1-\ell}\bar{\partial}^{\ell-1}\bar{\sigma}(u,z) \right), \\
\delta_{\tau_{n-1}}\sigma(u,z) &= \sum_{\ell=1}^{n} \frac{\ell}{(n-\ell)!} (\bar{\partial}^{s-\ell}\tau_{n-1})\partial_u^{-1}\left( u^{n-\ell}\partial_u^{3-\ell}\bar{\partial}^{\ell-1}\sigma(u,z) \right),
\end{aligned}
\tag{58}
$$

with coincides with the twistor actions of Proposition 1 with generators $\tau_{n-1} = g_{\dot{\alpha}(n)}\bar{\lambda}^{\dot{\alpha}(n)}$. In [49], a direct proof that this action forms a representation of the Schouten-Nijenhuis algebra of multivector fields, [70]

$$
[\tau_{n-1}, \tau'_{m-1}] = m\tau'_{m-1}\bar{\partial}\tau_{n-1} - n\tau_{n-1}\bar{\partial}\tau'_{m-1},
\tag{59}
$$

was given under the condition that $\bar{\partial}^{n+1}\tau_{n-1} = 0$, which is satisfied by the generators $\tau_{n-1} = g_{\dot{\alpha}(n)}\bar{\lambda}^{\dot{\alpha}(n)}$. This condition corresponds to the vanishing of the integrated soft charge. This result confirms the conclusion of Proposition 2 derived from the twistor action. In [33,49], it was also shown that relaxing the condition $\bar{\partial}^{n+1}\tau_{n-1} = 0$ is possible at the price of having a non-linear action of the Schouten-Nijenhuis algebra on the asymptotic phase space. We leave the discussion on the meaning of this nonlinear extension from the twistor space perspective for future work.

# 5 Details of the proof

This section contains the proof of Proposition 1, following the different steps as described in Fig. 1.

## 5.1 Positive helicity

We will evaluate the action of the algebra on plane waves,

$$\bar{\sigma}(u,\lambda) = \mp\frac{i\kappa}{8\pi^2}e^{\mp i\omega_0 u}\delta(\langle\lambda w\rangle) \in \mathcal{C}^\infty_{(-\frac{1}{2},\frac{3}{2})}(\mathscr{I}_{\mathcal{A}}), \tag{60}$$

by linearity of the representation it will extend to any Carrollian field (12) admitting a Fourier transform.

The first step $(i)$ consists in uplifting the self-dual Carrollian field (60) to twistor space. One easily obtains the twistor representative $\mathbf{h} = h(u = \mu^{\dot{\alpha}}\bar{\lambda}_{\dot{\alpha}}, \lambda)D\bar{\lambda}$ with

$$h(\mu^{\dot{\alpha}}\bar{\lambda}_{\dot{\alpha}}, \lambda) = \frac{\kappa}{8\pi^2\omega_0}e^{\mp i\omega_0(\mu^{\dot{\alpha}}\bar{\lambda}_{\dot{\alpha}})}\delta(\langle\lambda w\rangle). \tag{61}$$

The action of the $w_{1+\infty}$ symmetries on $\mathbf{h}$ is given by (36)

$$\delta\mathbf{h} = \{g,\mathbf{h}\} = \left(\frac{\partial g}{\partial\mu^{\dot{\alpha}}}\right)\epsilon^{\dot{\alpha}\dot{\beta}}\frac{\partial h}{\partial\mu^{\dot{\beta}}}D\bar{\lambda}, \tag{62}$$

with the generators $g$ given in (31). We thus get, at fixed $n$,

$$\begin{aligned}
\delta\mathbf{h}(\mu,\lambda) &= \frac{\mp i\kappa}{8\pi^2}\left(\frac{\partial g}{\partial\mu^{\dot{\alpha}_1}}\right)\bar{\lambda}^{\dot{\alpha}_1}e^{\mp i\omega_0(\mu^{\dot{\alpha}}\bar{\lambda}_{\dot{\alpha}})}\delta(\langle\lambda w\rangle)[\bar{\lambda}d\bar{\lambda}] \\
&= \frac{\mp i\kappa n}{8\pi^2}G^{(n)}(\mu,\lambda)\delta(\langle\lambda w\rangle)[\bar{\lambda}d\bar{\lambda}],
\end{aligned} \tag{63}$$

where

$$G^{(n)}(\mu,\lambda) := g_{\dot{\alpha}_1\ldots\dot{\alpha}_n}\bar{\lambda}^{\dot{\alpha}_1}\mu^{\dot{\alpha}_2}\ldots\mu^{\dot{\alpha}_n}e^{\mp i\omega_0(\mu^{\dot{\alpha}}\bar{\lambda}_{\dot{\alpha}})}. \tag{64}$$

We now implement the second step $(ii)$ by plugging the transformed twistor representative $\mathbf{h}$ into the Penrose transform. This leads to

$$\begin{aligned}
\delta\Phi_{\alpha\dot{\alpha}\beta\dot{\beta}}(x) &= \frac{1}{2\pi i}\int_{\mathbb{CP}^1}\langle\zeta d\zeta\rangle \wedge \frac{\iota_\alpha\iota_\beta}{\langle\iota\zeta\rangle^2}\frac{\partial^2\delta\mathbf{h}}{\partial\mu^{\dot{\alpha}}\partial\mu^{\dot{\beta}}}(\mu^{\dot{\alpha}} = x^{\alpha\dot{\alpha}}\zeta_\alpha, \zeta_\alpha) \\
&= \mp\frac{\kappa n}{16\pi^3}\int_{\mathbb{CP}^1}\langle\zeta d\zeta\rangle \wedge [\bar{\zeta}d\bar{\zeta}]\delta(\langle\zeta w\rangle)\frac{\iota_\alpha\iota_\beta}{\langle\iota\zeta\rangle^2}\frac{\partial^2 G^{(n)}}{\partial\mu^{\dot{\alpha}}\partial\mu^{\dot{\beta}}}(\mu^{\dot{\alpha}} = x^{\alpha\dot{\alpha}}\zeta_\alpha, \zeta_\alpha) \\
&= \pm\frac{i\kappa n}{8\pi^3}\frac{\iota_\alpha\iota_\beta}{\langle\iota w\rangle^2}\frac{\partial^2 G^{(n)}}{\partial\mu^{\dot{\alpha}}\partial\mu^{\dot{\beta}}}(\mu^{\dot{\alpha}} = x^{\alpha\dot{\alpha}}w_\alpha, w_\alpha),
\end{aligned} \tag{65}$$

where

$$\frac{\partial^2 G^{(n)}}{\partial \mu^{\dot\alpha}\partial \mu^{\dot\beta}}(\mu,w) = \Big(-(\omega_0)^2 g_{\dot\alpha_1\ldots\dot\alpha_n}\bar{w}^{\dot\alpha_1}\mu^{\dot\alpha_2}\ldots\mu^{\dot\alpha_n}\bar{w}_{\dot\alpha}\bar{w}_{\dot\beta} \mp i\omega_0 2(n-1)\bar{w}_{(\dot\beta}g_{\dot\alpha)\dot\alpha_1\ldots\dot\alpha_{n-1}}\bar{w}^{\dot\alpha_1}\mu^{\dot\alpha_2}\ldots\mu^{\dot\alpha_{n-1}}$$
$$+ (n-1)(n-2)g_{\dot\alpha\dot\beta\dot\alpha_1\ldots\dot\alpha_{n-2}}\bar{w}^{\dot\alpha_1}\mu^{\dot\alpha_2}\ldots\mu^{\dot\alpha_{n-2}}\Big)e^{\mp i\omega_0(\mu^{\dot\alpha}\bar{w}_{\dot\alpha})}, \tag{66}$$

and where we used the $\delta$-function normalisation $\int \frac{i}{2}d\zeta d\bar\zeta\,\delta(\zeta) = 1$. Note that in the above equation the generators $g_{\dot\alpha(n)}$ really are functions of $w$, which correspond to the direction of the plane wave, and not functions of $z$ which are the Bondi coordinates. Contracting (65) with (9), we obtain the transformed bulk field

$$\delta\Phi_{\bar z\bar z}(x) = \pm r^2\frac{i\kappa n}{8\pi^3}\frac{\langle \iota\lambda\rangle^2}{\langle \iota w\rangle^2}\bar n^{\dot\alpha}\bar n^{\dot\beta}\frac{\partial^2 G^{(n)}}{\partial\mu^{\dot\alpha}\partial\mu^{\dot\beta}}(\mu^{\dot\alpha} = x^{\alpha\dot\alpha}w_\alpha, w_\alpha). \tag{67}$$

We will split the computation according to the three different terms that appear in (66), namely

$$\delta\Phi_{\bar z\bar z}(x) := F_1(x) + F_2(x) + F_3(x), \tag{68}$$

with

$$F_1(x) = \pm r^2\frac{i\kappa n}{8\pi^3}\frac{\langle \iota\lambda\rangle^2}{\langle \iota w\rangle^2}\bar n^{\dot\alpha}\bar n^{\dot\beta}\Big(-(\omega_0)^2 g_{\dot\alpha_1\ldots\dot\alpha_n}\bar{w}^{\dot\alpha_1}\mu^{\dot\alpha_2}\ldots\mu^{\dot\alpha_n}\bar{w}_{\dot\alpha}\bar{w}_{\dot\beta}\Big)e^{\mp i\omega_0(\mu^{\dot\alpha}\bar{w}_{\dot\alpha})}\Big|_{\mu^{\dot\alpha}=x^{\alpha\dot\alpha}w_\alpha}$$
$$= -\frac{\langle \iota\lambda\rangle^2}{\langle \iota w\rangle^2}\Big((\omega_0)^2 g_{\dot\alpha(n)}\bar{w}^{\dot\alpha}\mu^{\dot\alpha(n-1)}\Big)\Big(\pm r^2\frac{i\kappa n}{8\pi^3}e^{\mp i\omega_0(\mu^{\dot\alpha}\bar{w}_{\dot\alpha})}\Big)\Big|_{\mu^{\dot\alpha}=x^{\alpha\dot\alpha}w_\alpha}, \tag{69}$$

where we recall that we use the notation $\dot\alpha(n) := (\dot\alpha_1\ldots\dot\alpha_n)$ for $n$ symmetrized indices,

$$F_2(x) = \pm r^2\frac{i\kappa n}{8\pi^3}\frac{\langle \iota\lambda\rangle^2}{\langle \iota w\rangle^2}\bar n^{\dot\alpha}\bar n^{\dot\beta}\Big(\mp i\omega_0 2(n-1)\bar{w}_{(\dot\beta}g_{\dot\alpha)\dot\alpha_1\ldots\dot\alpha_{n-1}}\bar{w}^{\dot\alpha_1}\mu^{\dot\alpha_2}\ldots\mu^{\dot\alpha_{n-1}}\Big)e^{\mp i\omega_0(\mu^{\dot\alpha}\bar{w}_{\dot\alpha})}\Big|_{\mu^{\dot\alpha}=x^{\alpha\dot\alpha}w_\alpha}$$
$$= \mp 2i(n-1)\frac{\langle \iota\lambda\rangle^2}{\langle \iota w\rangle^2}\Big(\omega_0\, g_{\dot\alpha(n)}\bar n^{\dot\alpha}\bar{w}^{\dot\alpha}\mu^{\dot\alpha(n-2)}\Big)\Big(\pm r^2\frac{i\kappa n}{8\pi^3}e^{\mp i\omega_0(\mu^{\dot\alpha}\bar{w}_{\dot\alpha})}\Big)\Big|_{\mu^{\dot\alpha}=x^{\alpha\dot\alpha}w_\alpha}, \tag{70}$$

and

$$F_3(x) = \pm r^2\frac{i\kappa n}{8\pi^3}\frac{\langle \iota\lambda\rangle^2}{\langle \iota w\rangle^2}\bar n^{\dot\alpha}\bar n^{\dot\beta}\Big((n-1)(n-2)g_{\dot\alpha\dot\beta\dot\alpha_1\ldots\dot\alpha_{n-2}}\bar{w}^{\dot\alpha_1}\mu^{\dot\alpha_2}\ldots\mu^{\dot\alpha_{n-2}}\Big)e^{\mp i\omega_0(\mu^{\dot\alpha}\bar{w}_{\dot\alpha})}\Big|_{\mu^{\dot\alpha}=x^{\alpha\dot\alpha}w_\alpha}$$
$$= (n-1)(n-2)\frac{\langle \iota\lambda\rangle^2}{\langle \iota w\rangle^2}\Big(g_{\dot\alpha(n)}\bar n^{\dot\alpha(2)}\bar{w}^{\dot\alpha}\mu^{\dot\alpha(n-3)}\Big)\Big(\pm r^2\frac{i\kappa n}{8\pi^3}e^{\mp i\omega_0(\mu^{\dot\alpha}\bar{w}_{\dot\alpha})}\Big)\Big|_{\mu^{\dot\alpha}=x^{\alpha\dot\alpha}w_\alpha}. \tag{71}$$

We will first focus on working out the first term (69), which reads

$$F_1(x) = -\frac{\langle \iota\lambda\rangle^2}{\langle \iota w\rangle^2}\Big((\omega_0)^2 g_{\dot\alpha(n)}\bar{w}^{\dot\alpha}x^{\dot\alpha(n-1)\alpha(n-1)}w_{\alpha(n-1)}\Big)\Big(\pm r^2\frac{i\kappa n}{8\pi^3}e^{\mp i\omega_0(x^{\alpha\dot\alpha}w_\alpha\bar{w}_{\dot\alpha})}\Big). \tag{72}$$

Remembering that

$$x^{\alpha\dot\alpha}w_\alpha = (u\,n^\alpha\bar n^{\dot\alpha} + r\,\lambda^\alpha\bar\lambda^{\dot\alpha})w_\alpha$$
$$= r\bar\lambda^{\dot\alpha}\langle\lambda w\rangle + u\bar n^{\dot\alpha}, \tag{73}$$

one finds the identity

$$g_{\dot\alpha(n)}x^{\dot\alpha(n-1)\alpha(n-1)}w_{\alpha(n-1)} = g_{\dot\alpha(n)}\big(r\bar\lambda^{\dot\alpha_1}\langle\lambda w\rangle + u\bar n^{\dot\alpha_1}\big)\ldots\big(r\bar\lambda^{\dot\alpha_{n-1}}\langle\lambda w\rangle + u\bar n^{\dot\alpha_{n-1}}\big)$$
$$= g_{\dot\alpha(n)}\sum_{\ell=0}^{n-1}\binom{n-1}{\ell}\big(r\langle\lambda w\rangle\big)^\ell\bar\lambda^{\dot\alpha(\ell)}u^{n-1-\ell}\bar n^{\dot\alpha(n-1-\ell)}. \tag{74}$$

We now arrive at a crucial step in the proof (step $(iii)$), which consists of performing the large-$r$ expansion in order to obtain the transformed field at $\mathscr{I}$. At face value, it seems from the presence of $r^{\ell}$ terms in expression (74) that (72) will render an expression which dramatically blows up at $\mathscr{I}$. This is however not the case, due to a mechanism we will now detail.

As $r \to \infty$, the plane waves admit to be written as[11]

$$
\begin{aligned}
e^{\mp i\omega_0(x^{\alpha\dot\alpha}w_\alpha\bar w_{\dot\alpha})} &= e^{\mp i\omega_0(u+r|z-w|^2)} \\
&= \mp i\pi \frac{e^{\mp i\omega_0 u}}{r\omega_0}\left(1 + \frac{1}{\pm i\omega_0 r}\partial\bar\partial + \frac{1}{2(\pm i\omega_0 r)^2}(\partial\bar\partial)^2 + \dots\right)\delta(\langle\lambda w\rangle) \\
&= \mp i\pi \frac{e^{\mp i\omega_0 u}}{r\omega_0}\sum_{k=0}^{\infty}\frac{1}{k!}\left(\frac{r^{-1}}{\pm i\omega_0}\partial\bar\partial\right)^k\delta(\langle\lambda w\rangle),
\end{aligned}
\tag{75}
$$

and hence

$$
\pm r^2 \frac{in\kappa}{8\pi^3}e^{\mp i\omega_0(x^{\alpha\dot\alpha}w_\alpha\bar w_{\dot\alpha})} = \frac{rn\kappa}{8\pi^2\omega_0}e^{\mp i\omega_0 u}\sum_{k=0}^{\infty}\frac{1}{k!}\left(\frac{r^{-1}}{\pm i\omega_0}\partial\bar\partial\right)^k\delta(\langle\lambda w\rangle).
\tag{76}
$$

Plugging (74) and (76) into (72), we arrive at

$$
F_1(x) = -\frac{rn\kappa\omega_0}{8\pi^2}e^{\mp i\omega_0 u}\frac{\langle\iota\lambda\rangle^2}{\langle\iota w\rangle^2}
\tag{77}
$$

$$
\times\, g_{\dot\alpha(n)}\bar w^{\dot\alpha}\sum_{\ell=0}^{n-1}\sum_{k=0}^{\infty}\left(\binom{n-1}{\ell}(r\langle\lambda w\rangle)^{\ell}\,\bar\lambda^{\dot\alpha(\ell)}u^{n-1-\ell}\bar n^{\dot\alpha(n-1-\ell)}\right)\frac{1}{k!}\left(\frac{r^{-1}}{\pm i\omega_0}\partial\bar\partial\right)^k\delta(\langle\lambda w\rangle).
$$

Therefore, thanks to the distributional identity

$$
\langle\lambda w\rangle^m\partial^n\delta(\langle\lambda w\rangle) = \delta_{m,n}(-1)^n n!\,\delta(\langle\lambda w\rangle), \qquad \forall\, m \geq n,
\tag{78}
$$

we see that each overleading term in $r$ vanishes! Moreover, all factors neatly combine to give a leading $\mathcal{O}(r)$ term free from the residual gauge ambiguity related to $\iota^\alpha$, namely

$$
F_1(x) = \frac{rn\kappa\omega_0}{8\pi^2}e^{\mp i\omega_0 u}\,g_{\dot\alpha(n)}\bar w^{\dot\alpha}\sum_{\ell=0}^{n-1}(-1)^{\ell+1}\binom{n-1}{\ell}\bar\lambda^{\dot\alpha(\ell)}\bar n^{\dot\alpha(n-1-\ell)}u^{n-1-\ell}\left(\frac{\bar\partial}{\pm i\omega_0}\right)^{\ell}\delta(\langle\lambda w\rangle) + \mathcal{O}(r^0).
\tag{79}
$$

We can now rewrite $F_1$ by recalling the definition of the plane waves (60); using

$$
(\pm i\omega_0)^m e^{\mp i\omega_0 u} = (-\partial_u)^m e^{\mp i\omega_0 u},
\tag{80}
$$

we get

$$
\begin{aligned}
F_1(x) &= \pm irn\omega_0 g_{\dot\alpha(n)}\bar w^{\dot\alpha}\left[\sum_{\ell=0}^{n-1}(-1)^{\ell+1}\binom{n-1}{\ell}\bar\lambda^{\dot\alpha(\ell)}\bar n^{\dot\alpha(n-1-\ell)}u^{n-1-\ell}\left(\frac{\bar\partial}{\pm i\omega_0}\right)^{\ell}\right] \\
&\quad\times\left(\frac{\mp i\kappa}{8\pi^2}e^{\mp i\omega_0 u}\delta(\langle\lambda w\rangle)\right) + \mathcal{O}(r^0) \\
&= rn g_{\dot\alpha(n)}\bar w^{\dot\alpha}\left[\sum_{\ell=0}^{n-1}\binom{n-1}{\ell}\bar\lambda^{\dot\alpha(\ell)}\bar n^{\dot\alpha(n-1-\ell)}\,u^{n-1-\ell}\,\bar\partial^{\ell}(\partial_u)^{1-\ell}\right]\bar\sigma + \mathcal{O}(r^0).
\end{aligned}
\tag{81}
$$

---

[11]This expression can be derived by looking for the unique solution of the wave equation in Bondi coordinates $\left(-\frac{1}{r}\partial_u - \partial_u\partial_r + \frac{1}{r^2}\partial_z\partial_{\bar z}\right)\phi = 0$ which is the form $e^{\mp i\omega_0 u}\psi(r,z,\bar z)$ and satisfies the asymptotic boundary condition $e^{\mp i\omega_0(x^{\alpha\dot\alpha}w_\alpha\bar w_{\dot\alpha})} = \mp\frac{\pi i}{\omega_0 r}e^{\mp i\omega_0 u}\delta(z-\zeta) + \mathcal{O}(r^{-2})$. See also [71] for similar asymptotic expressions.

We now make use of the following identity

$$\bar{w}^{\dot\alpha}(\bar\partial)^m \delta(\langle \lambda w\rangle) = \bar\lambda^{\dot\alpha}(\bar\partial)^m \delta(\langle \lambda w\rangle) + m\bar\partial \bar\lambda^{\dot\alpha}(\bar\partial)^{m-1}\delta(\langle \lambda w\rangle), \tag{82}$$

and obtain

$$
\begin{aligned}
F_1(x) &= rng_{\dot\alpha(n)}\Bigg[ \sum_{\ell=0}^{n-1} \binom{n-1}{\ell} \bar\lambda^{\dot\alpha(\ell+1)} \bar{n}^{\dot\alpha(n-1-\ell)} u^{n-1-\ell} \, \bar\partial^\ell (\partial_u)^{1-\ell} \\
&\qquad\qquad + \sum_{\ell=0}^{n-1} \ell\binom{n-1}{\ell} \bar\lambda^{\dot\alpha(\ell)} \bar{n}^{\dot\alpha(n-\ell)} u^{n-1-\ell} \, \bar\partial^{\ell-1}(\partial_u)^{1-\ell} \Bigg]\bar\sigma + \mathcal{O}(r^0) \\
&= rng_{\dot\alpha(n)}\Bigg[ \sum_{\ell=1}^{n} \binom{n-1}{\ell-1} \bar\lambda^{\dot\alpha(\ell)} \bar{n}^{\dot\alpha(n-\ell)} u^{n-\ell} \, \bar\partial^{\ell-1}(\partial_u)^{2-\ell} \\
&\qquad\qquad + \sum_{\ell=1}^{n-1} \ell\binom{n-1}{\ell} \bar\lambda^{\dot\alpha(\ell)} \bar{n}^{\dot\alpha(n-\ell)} u^{n-1-\ell} \, \bar\partial^{\ell-1}(\partial_u)^{1-\ell} \Bigg]\bar\sigma + \mathcal{O}(r^0) \\
&= rng_{\dot\alpha(n)}\Bigg[ \sum_{\ell=1}^{n} \bar\lambda^{\dot\alpha(\ell)} \bar{n}^{\dot\alpha(n-\ell)} \binom{n-1}{\ell-1}\Big(u^{n-\ell}(\partial_u)^{2-\ell} + (n-\ell)u^{n-1-\ell}(\partial_u)^{1-\ell}\Big)\bar\partial^{\ell-1}\Bigg]\bar\sigma + \mathcal{O}(r^0) \\
&= rng_{\dot\alpha(n)} \sum_{\ell=1}^{n} \bar\lambda^{\dot\alpha(\ell)} \bar{n}^{\dot\alpha(n-\ell)} \binom{n-1}{\ell-1} \partial_u\Big(u^{n-\ell}(\partial_u)^{1-\ell}\,\bar\partial^{\ell-1}\bar\sigma\Big) + \mathcal{O}(r^0),
\end{aligned}
\tag{83}
$$

where we used $\binom{n-1}{\ell} = \frac{n-\ell}{\ell}\binom{n-1}{\ell-1}$ in the third equality. Finally, using that

$$g_{\dot\alpha(n)} \bar\lambda^{\dot\alpha(\ell)} n^{\dot\alpha(n-\ell)} = g_{\dot\alpha(n)} \frac{\ell!}{n!} \bar\partial^{n-\ell}\left(\bar\lambda^{\dot\alpha(n)}\right), \tag{84}$$

we end up with

$$
\begin{aligned}
F_1(x) &= rng_{\dot\alpha(n)} \sum_{\ell=1}^{n} \bar\partial^{n-\ell}\left(\bar\lambda^{\dot\alpha(n)}\right) \frac{\ell!}{n!}\binom{n-1}{\ell-1} \partial_u\Big(u^{n-\ell}(\partial_u)^{1-\ell}\,\bar\partial^{\ell-1}\bar\sigma\Big) + \mathcal{O}(r^0) \\
&= rg_{\dot\alpha(n)} \sum_{\ell=1}^{n} \bar\partial^{n-\ell}\left(\bar\lambda^{\dot\alpha(n)}\right) \frac{\ell}{(n-\ell)!} \partial_u\Big(u^{n-\ell}(\partial_u)^{1-\ell}\,\bar\partial^{\ell-1}\bar\sigma\Big) + \mathcal{O}(r^0).
\end{aligned}
\tag{85}
$$

The computation for the two remaining terms will follow along very similar lines. The second term (70) reads

$$F_2(x) = \mp 2i(n-1)\frac{\langle \iota\lambda\rangle^2}{\langle \iota w\rangle^2}\left(\omega_0\, g_{\dot\alpha(n)} \bar{n}^{\dot\alpha}\bar{w}^{\dot\alpha} x^{\dot\alpha(n-2)\alpha(n-2)} w_{\alpha(n-2)}\right)\left(\pm r^2 \frac{i\kappa n}{8\pi^3} e^{\mp i\omega_0(x^{\alpha\dot\alpha} w_\alpha \bar{w}_{\dot\alpha})}\right). \tag{86}$$

Using again (76) together with the identity

$$g_{\dot\alpha(n)} x^{\dot\alpha(n-2)\alpha(n-2)} w_{\alpha(n-2)} = g_{\dot\alpha(n)} \sum_{\ell=0}^{n-2} \binom{n-2}{\ell}(r\langle \lambda w\rangle)^\ell \,\bar\lambda^{\dot\alpha(\ell)} u^{n-2-\ell} \bar{n}^{\dot\alpha(n-2-\ell)}, \tag{87}$$

we obtain

$$
\begin{aligned}
F_2(x) &= \mp \frac{n(n-1)ir\kappa}{4\pi^2} e^{\mp i\omega_0 u}\frac{\langle \iota\lambda\rangle^2}{\langle \iota w\rangle^2}\left(g_{\dot\alpha(n)} \bar{n}^{\dot\alpha}\bar{w}^{\dot\alpha} \sum_{\ell=0}^{n-2} \binom{n-2}{\ell}(r\langle \lambda w\rangle)^\ell \,\bar\lambda^{\dot\alpha(\ell)} u^{n-2-\ell}\right)\bar{n}^{\dot\alpha(n-2-\ell)}\Big) \\
&\quad \times \sum_{k=0}^{\infty} \frac{1}{k!}\left(\frac{r^{-1}}{\pm i\omega_0}\partial\bar\partial\right)^k \delta(\langle \lambda w\rangle).
\end{aligned}
\tag{88}
$$

All overleading terms in $r$ and residual gauge factors in (88) again disappear thanks to the identity (78) and we are left with

$$F_2 = \frac{\mp n(n-1)ir\kappa}{4\pi^2}e^{\mp i\omega_0 u}g_{\dot\alpha(n)}\bar{w}^{\dot\alpha}\sum_{\ell=0}^{n-2}(-1)^\ell\binom{n-2}{\ell}\bar\lambda^{\dot\alpha(\ell)}\bar{n}^{\dot\alpha(n-1-\ell)}u^{n-2-\ell}\left(\frac{\bar\partial}{\pm i\omega_0}\right)^\ell\delta(\langle\lambda w\rangle)+\mathcal{O}(r^0)$$

$$= 2n(n-1)r\,g_{\dot\alpha(n)}\bar{w}^{\dot\alpha}\left[\sum_{\ell=0}^{n-2}\binom{n-2}{\ell}\bar\lambda^{\dot\alpha(\ell)}\bar{n}^{\dot\alpha(n-1-\ell)}u^{n-2-\ell}\bar\partial^\ell\partial_u^{-\ell}\right]\bar\sigma+\mathcal{O}(r^0), \tag{89}$$

where we have reinstated $\bar\sigma$ as given in (60). We can now make use of the identity (82) to write

$$F_2(x) = 2n(n-1)rg_{\dot\alpha(n)}\left[\sum_{\ell=0}^{n-2}\binom{n-2}{\ell}\bar\lambda^{\dot\alpha(\ell+1)}\bar{n}^{\dot\alpha(n-1-\ell)}u^{n-2-\ell}\bar\partial^\ell\partial_u^{-\ell}\right.$$

$$\left.+\sum_{\ell=0}^{n-2}\ell\binom{n-2}{\ell}\bar\lambda^{\dot\alpha(\ell)}\bar{n}^{\dot\alpha(n-\ell)}u^{n-2-\ell}\bar\partial^{\ell-1}\partial_u^{-\ell}\right]\bar\sigma+\mathcal{O}(r^0)$$

$$= 2n(n-1)rg_{\dot\alpha(n)}\left[\sum_{\ell=1}^{n-1}\binom{n-2}{\ell-1}\bar\lambda^{\dot\alpha(\ell)}\bar{n}^{\dot\alpha(n-\ell)}u^{n-1-\ell}\bar\partial^{\ell-1}\partial_u^{1-\ell}\right. \tag{90}$$

$$\left.+\sum_{\ell=1}^{n-2}\ell\binom{n-2}{\ell}\bar\lambda^{\dot\alpha(\ell)}\bar{n}^{\dot\alpha(n-\ell)}u^{n-2-\ell}\bar\partial^{\ell-1}\partial_u^{-\ell}\right]\bar\sigma+\mathcal{O}(r^0)$$

$$= 2n(n-1)rg_{\dot\alpha(n)}\left[\sum_{\ell=1}^{n-1}\bar\lambda^{\dot\alpha(\ell)}\bar{n}^{\dot\alpha(n-\ell)}\binom{n-2}{\ell-1}\left(u^{n-1-\ell}\partial_u^{1-\ell}+(n-1-\ell)u^{n-2-\ell}\partial_u^{-\ell}\right)\bar\partial^{\ell-1}\right]\bar\sigma$$

$$+\mathcal{O}(r^0)$$

$$= 2n(n-1)rg_{\dot\alpha(n)}\sum_{\ell=1}^{n-1}\bar\lambda^{\dot\alpha(\ell)}\bar{n}^{\dot\alpha(n-\ell)}\binom{n-2}{\ell-1}\partial_u\left(u^{n-1-\ell}\partial_u^{-\ell}\bar\partial^{\ell-1}\bar\sigma\right)+\mathcal{O}(r^0),$$

where we used $\binom{n-2}{\ell}=\frac{(n-\ell-1)}{\ell}\binom{n-2}{\ell-1}$ in the third equality. Finally, using (84), we end up with

$$F_2(x) = 2n(n-1)rg_{\dot\alpha(n)}\sum_{\ell=1}^{n-1}\bar\partial^{n-\ell}\left(\bar\lambda^{\dot\alpha(n)}\right)\frac{\ell!}{n!}\binom{n-2}{\ell-1}\partial_u\left(u^{n-1-\ell}\partial_u^{-\ell}\bar\partial^{\ell-1}\bar\sigma\right)+\mathcal{O}(r^0)$$

$$= 2rg_{\dot\alpha(n)}\sum_{\ell=1}^{n-1}\bar\partial^{n-\ell}\left(\bar\lambda^{\dot\alpha(n)}\right)\frac{\ell}{(n-\ell-1)!}\partial_u\left(u^{n-1-\ell}\partial_u^{-\ell}\bar\partial^{\ell-1}\bar\sigma\right)+\mathcal{O}(r^0). \tag{91}$$

Finally, following the same procedure as described above for the last term (71), we get

$$F_3(x) = (n-1)(n-2)\frac{\langle\iota\lambda\rangle^2}{\langle\iota w\rangle^2}\left(g_{\dot\alpha(n)}\bar{n}^{\dot\alpha(2)}\bar{w}^{\dot\alpha}x^{\dot\alpha(n-3)\alpha(n-3)}w_{\alpha(n-3)}\right)\left(\pm r^2\frac{i\kappa n}{8\pi^3}e^{\mp i\omega_0(x^{\alpha\dot\alpha}w_\alpha\bar{w}_{\dot\alpha})}\right),$$

$$= \frac{n(n-1)(n-2)r\kappa}{8\pi^2\omega_0}e^{\mp i\omega_0 u}\frac{\langle\iota\lambda\rangle^2}{\langle\iota w\rangle^2}\left(g_{\dot\alpha(n)}\bar{n}^{\dot\alpha(2)}\bar{w}^{\dot\alpha}u^{n-3-\ell}\sum_{\ell=0}^{n-3}\binom{n-3}{\ell}(r\langle\lambda w\rangle)^\ell\bar\lambda^{\dot\alpha(\ell)}\bar{n}^{\dot\alpha(n-3-\ell)}\right)$$

$$\times\sum_{k=0}^{\infty}\frac{1}{k!}\left(\frac{r^{-1}}{\pm i\omega_0}\partial\bar\partial\right)^k\delta(\langle\lambda w\rangle) \tag{92}$$

$$= \frac{n(n-1)(n-2)r\kappa}{8\pi^2\omega_0}e^{\mp i\omega_0 u}g_{\dot\alpha(n)}\bar{w}^{\dot\alpha}\sum_{\ell=0}^{n-3}(-1)^\ell\binom{n-3}{\ell}\bar\lambda^{\dot\alpha(\ell)}\bar{n}^{\dot\alpha(n-1-\ell)}u^{n-3-\ell}\left(\frac{\bar\partial}{\pm i\omega_0}\right)^\ell\delta(\langle\lambda w\rangle)$$

$$+\mathcal{O}(r^0)$$

$$= n(n-1)(n-2)rg_{\dot\alpha(n)}\bar{w}^{\dot\alpha}\left[\sum_{\ell=0}^{n-3}\binom{n-3}{\ell}\bar\lambda^{\dot\alpha(\ell)}\bar{n}^{\dot\alpha(n-1-\ell)}u^{n-3-\ell}\bar\partial^\ell(\partial_u)^{-1-\ell}\right]\bar\sigma+\mathcal{O}(r^0).$$

The identity (82) then leads to

$$
\begin{aligned}
F_3(x) &= n(n-1)(n-2)r g_{\dot\alpha(n)}\Bigg[\sum_{\ell=0}^{n-3}\binom{n-3}{\ell}\bar\lambda^{\dot\alpha(\ell+1)}\bar n^{\dot\alpha(n-1-\ell)}u^{n-3-\ell}\bar\partial^\ell(\partial_u)^{-1-\ell}\\
&\qquad\qquad\qquad +\sum_{\ell=0}^{n-3}\ell\binom{n-3}{\ell}\bar\lambda^{\dot\alpha(\ell)}\bar n^{\dot\alpha(n-\ell)}u^{n-3-\ell}\bar\partial^{\ell-1}(\partial_u)^{-1-\ell}\Bigg]\bar\sigma+\mathcal{O}(r^0)\\
&= n(n-1)(n-2)r g_{\dot\alpha(n)}\Bigg[\sum_{\ell=1}^{n-2}\binom{n-3}{\ell-1}\bar\lambda^{\dot\alpha(\ell)}\bar n^{\dot\alpha(n-\ell)}u^{n-2-\ell}\bar\partial^{\ell-1}(\partial_u)^{-\ell}\\
&\qquad\qquad\qquad +\sum_{\ell=1}^{n-3}\ell\binom{n-3}{\ell}\bar\lambda^{\dot\alpha(\ell)}\bar n^{\dot\alpha(n-\ell)}u^{n-3-\ell}\bar\partial^{\ell-1}(\partial_u)^{-1-\ell}\Bigg]\bar\sigma+\mathcal{O}(r^0)\\
&= n(n-1)(n-2)r g_{\dot\alpha(n)}\sum_{\ell=1}^{n-2}\bar\lambda^{\dot\alpha(\ell)}\bar n^{\dot\alpha(n-\ell)}\binom{n-3}{\ell-1}\partial_u\Big(u^{n-2-\ell}(\partial_u)^{-\ell-1}\bar\partial^{\ell-1}\bar\sigma\Big)+\mathcal{O}(r^0),
\end{aligned}
$$
(93)

where we used $\binom{n-3}{\ell}=\frac{(n-\ell-2)}{\ell}\binom{n-3}{\ell-1}$ to get the last equality. Finally, using (84), we end up with

$$
\begin{aligned}
F_3(x) &= n(n-1)(n-2)r g_{\dot\alpha(n)}\sum_{\ell=1}^{n-2}\bar\partial^{n-\ell}\big(\bar\lambda^{\dot\alpha(n)}\big)\frac{\ell!}{n!}\binom{n-3}{\ell-1}\partial_u\Big(u^{n-2-\ell}(\partial_u)^{-\ell-1}\bar\partial^{\ell-1}\bar\sigma\Big)+\mathcal{O}(r^0)\\
&= r g_{\dot\alpha(n)}\sum_{\ell=1}^{n-2}\bar\partial^{n-\ell}\big(\bar\lambda^{\dot\alpha(n)}\big)\frac{\ell}{(n-\ell-2)!}\partial_u\Big(u^{n-2-\ell}(\partial_u)^{-\ell-1}\bar\partial^{\ell-1}\bar\sigma\Big)+\mathcal{O}(r^0).
\end{aligned}
$$
(94)

We are now ready to collect all three terms $F_i(x)$ ($i=1,2,3$) and read off the action on the shear. From (85), (91) and (94), we obtain the final expression

$$
\begin{aligned}
\delta\bar\sigma &= g_{\dot\alpha(n)}\sum_{\ell=1}^{n}\bar\partial^{n-\ell}\big(\bar\lambda^{\dot\alpha(n)}\big)\frac{\ell}{(n-\ell)!}\partial_u\Big(u^{n-\ell}(\partial_u)^{1-\ell}\ \bar\partial^{\ell-1}\bar\sigma\Big)\\
&\quad +2g_{\dot\alpha(n)}\sum_{\ell=1}^{n-1}\bar\partial^{n-\ell}\big(\bar\lambda^{\dot\alpha(n)}\big)\frac{\ell}{(n-\ell-1)!}\partial_u\Big(u^{n-1-\ell}\partial_u^{-\ell}\bar\partial^{\ell-1}\bar\sigma\Big)\\
&\quad +g_{\dot\alpha(n)}\sum_{\ell=1}^{n-2}\bar\partial^{n-\ell}\big(\bar\lambda^{\dot\alpha(n)}\big)\frac{\ell}{(n-\ell-2)!}\partial_u\Big(u^{n-2-\ell}(\partial_u)^{-\ell-1}\bar\partial^{\ell-1}\bar\sigma\Big)\\
&= g_{\dot\alpha(n)}\sum_{\ell=1}^{n}\bar\partial^{n-\ell}\big(\bar\lambda^{\dot\alpha(n)}\big)\frac{\ell}{(n-\ell)!}\\
&\quad\times\partial_u\Big[\Big(u^{n-\ell}(\partial_u)^{1-\ell}\ +2(n-\ell)u^{n-1-\ell}\partial_u^{-\ell}+(n-\ell)(n-\ell-1)u^{n-2-\ell}(\partial_u)^{-\ell-1}\Big)\bar\partial^{\ell-1}\bar\sigma\Big]\\
&= g_{\dot\alpha(n)}\sum_{\ell=1}^{n}\bar\partial^{n-\ell}\big(\bar\lambda^{\dot\alpha(n)}\big)\frac{\ell}{(n-\ell)!}\partial_u^3\Big(u^{n-\ell}(\partial_u)^{-\ell-1}\bar\partial^{\ell-1}\bar\sigma\Big).
\end{aligned}
$$
(95)

We now need to remember that $g_{\dot\alpha(n)}$ here is a function of $w$ and can therefore freely move through the derivatives to give

$$
\delta\bar\sigma=\sum_{\ell=1}^{n}\bar\partial^{n-\ell}\big(\bar\lambda^{\dot\alpha(n)}\big)\frac{\ell}{(n-\ell)!}\partial_u^3\Big(u^{n-\ell}(\partial_u)^{-\ell-1}\bar\partial^{\ell-1}\big(g_{\dot\alpha(n)}\bar\sigma\big)\Big).
$$
(96)

In the above formula, the generator $g$ can now be taken to be a function of $z$ (since $\bar{\sigma}$ given in (60) is proportional to a Dirac $\delta$-function). Since, for any integer $k$, $\bar{\partial}^k g$ is only non-zero at $z = 0, z = \infty$, we can write (remembering (60) that the plane wave only has support on $\mathcal{A}$)

$$\delta\bar{\sigma} = \sum_{\ell=1}^{n} \bar{\partial}^{n-\ell}\left(g_{\dot{\alpha}(n)}\bar{\lambda}^{\dot{\alpha}(n)}\right)\frac{\ell}{(n-\ell)!}\partial_u^3\left(u^{n-\ell}(\partial_u)^{-\ell-1}\bar{\partial}^{\ell-1}\bar{\sigma}\right). \tag{97}$$

The formula finally extends to any Carrollian field $\bar{\sigma} \in \mathcal{C}^{\infty}_{(-\frac{1}{2},\frac{3}{2})}(\mathscr{I}_{\mathcal{A}})$ by linear combinations of plane waves. This concludes the derivation the first equality of Proposition 1.

## 5.2 Negative helicity

For the opposite helicity, we consider the plane waves

$$\sigma(u,\lambda) = \pm\frac{i\kappa}{8\pi^2}e^{\pm i\omega_0 u}\delta(\langle\lambda w\rangle) \in \mathcal{C}^{\infty}_{(\frac{3}{2},-\frac{1}{2})}(\mathscr{I}_{\mathcal{A}}). \tag{98}$$

The twistor representative is $\widetilde{\mathbf{h}} = \widetilde{h}\left(\mu^{\dot{\alpha}}\bar{\lambda}_{\dot{\alpha}},\lambda\right)D\bar{\lambda} \in \Omega^{0,1}(\mathbb{PT},\mathcal{O}(-6))$ with (19)

$$\widetilde{h} = \frac{\kappa\omega_0^3}{8\pi^2}e^{\pm i\omega_0(\mu^{\dot{\alpha}}\bar{\lambda}_{\dot{\alpha}})}\delta(\langle\lambda w\rangle). \tag{99}$$

The action of the $w_{1+\infty}$ symmetries on $\widetilde{\mathbf{h}}$ is given by (36)

$$\delta\widetilde{\mathbf{h}} = \{g,\widetilde{\mathbf{h}}\} = \left(\frac{\partial g}{\partial\mu^{\dot{\alpha}}}\right)\epsilon^{\dot{\alpha}\dot{\beta}}\frac{\partial\widetilde{h}}{\partial\mu^{\dot{\beta}}}D\bar{\lambda}, \tag{100}$$

and we get, at fixed $n$,

$$\begin{aligned}\delta\widetilde{\mathbf{h}}(\mu,\lambda) &= \frac{\pm i\kappa\omega_0^4}{8\pi^2}\left(\frac{\partial g}{\partial\mu^{\dot{\alpha}}}\right)\bar{\lambda}^{\dot{\alpha}}e^{\pm i\omega_0(\mu^{\dot{\alpha}}\bar{\lambda}_{\dot{\alpha}})}\delta(\langle\lambda w\rangle)D\bar{\lambda}\\ &= \frac{\pm i\kappa n\omega_0^4}{8\pi^2}g_{\dot{\alpha}_1\dots\dot{\alpha}_n}\bar{\lambda}^{\dot{\alpha}_1}\mu^{\dot{\alpha}_2}\dots\mu^{\dot{\alpha}_n}e^{\pm i\omega_0(\mu^{\dot{\alpha}}\bar{\lambda}_{\dot{\alpha}})}\delta(\langle\lambda w\rangle)D\bar{\lambda}.\end{aligned} \tag{101}$$

We can now plug the transformed twistor representative into expression (26) leading to the Weyl tensor:

$$\begin{aligned}\delta\overline{\Psi}_{\alpha\dot{\alpha}\beta\dot{\beta}}(x) &= \frac{i}{2\pi}\int_{\mathbb{CP}^1}\langle\zeta d\zeta\rangle\zeta_\alpha\zeta_\beta\zeta_\gamma\zeta_\delta\,\delta\widetilde{\mathbf{h}}(\mu^{\dot{\alpha}} = x^{\alpha\dot{\alpha}}\zeta_\alpha,\zeta_\alpha)\\ &= \mp\frac{\kappa n\omega_0^4}{16\pi^3}\int_{\mathbb{CP}^1}d\zeta\wedge d\bar{\zeta}\,\zeta_\alpha\zeta_\beta\zeta_\gamma\zeta_\delta\,g_{\dot{\alpha}_1\dots\dot{\alpha}_n}\bar{\zeta}^{\dot{\alpha}_1}\mu^{\dot{\alpha}_2}\dots\mu^{\dot{\alpha}_n}e^{\pm i\omega_0(\mu^{\dot{\alpha}}\bar{\zeta}_{\dot{\alpha}})}\delta(\langle\zeta w\rangle)\Big|_{\mu^{\dot{\alpha}}=x^{\alpha\dot{\alpha}}w_\alpha}\\ &= \pm\frac{i\kappa n\omega_0^4}{8\pi^3}w_\alpha w_\beta w_\gamma w_\delta\,g_{\dot{\alpha}_1\dots\dot{\alpha}_n}\bar{w}^{\dot{\alpha}_1}\mu^{\dot{\alpha}_2}\dots\mu^{\dot{\alpha}_n}e^{\pm i\omega_0(\mu^{\dot{\alpha}}\bar{w}_{\dot{\alpha}})}\Big|_{\mu^{\dot{\alpha}}=x^{\alpha\dot{\alpha}}w_\alpha},\end{aligned} \tag{102}$$

where we integrated the $\delta$-function. This leads to

$$\delta\overline{\Psi}_4(x) = \pm\frac{i\kappa n\omega_0^4}{8\pi^3}g_{\dot{\alpha}(n)}\bar{w}^{\dot{\alpha}}x^{\dot{\alpha}(n-1)\alpha(n-1)}w_{\alpha(n-1)}e^{\pm i\omega_0(x^{\alpha\dot{\alpha}}w_\alpha\bar{w}_{\dot{\alpha}})}. \tag{103}$$

We already computed a very similar expression in the previous section; see (72). We can thus directly use the result (85) and read from (30) that

$$\begin{aligned}\delta\sigma &= -\partial_u^{-2}\lim_{r\to\infty}r\delta\overline{\Psi}_4(x)\\ &= \partial_u^{-2}\sum_{\ell=0}^{n}\bar{\partial}^{n-\ell}\left(\bar{\lambda}^{\dot{\alpha}(n)}\right)\frac{\ell}{(n-\ell)!}\partial_u\left(u^{n-\ell}(\partial_u)^{1-\ell}\bar{\partial}^{\ell-1}\left(g_{\dot{\alpha}(n)}\partial_u^2\sigma\right)\right)\\ &= \sum_{\ell=0}^{n}\bar{\partial}^{n-\ell}\left(\bar{\lambda}^{\dot{\alpha}(n)}\right)\frac{\ell}{(n-\ell)!}\partial_u^{-1}\left(u^{n-\ell}(\partial_u)^{3-\ell}\bar{\partial}^{\ell-1}\left(g_{\dot{\alpha}(n)}\sigma\right)\right).\end{aligned} \tag{104}$$

By the same argument that we used for the positive helicity case, we can replace the $w$ dependence of $g$ by a dependence on $z$ and are allowed to move it freely through the derivatives thanks to our requirement (98) on the support of $\sigma$, therefore

$$\delta\sigma = \sum_{\ell=0}^{n} \bar{\partial}^{n-\ell}\Big(g_{\dot{\alpha}(n)}\bar{\lambda}^{\dot{\alpha}(n)}\Big)\frac{\ell}{(n-\ell)!}\partial_u^{-1}\Big(u^{n-\ell}(\partial_u)^{3-\ell}\bar{\partial}^{\ell-1}\sigma\Big). \tag{105}$$

The formula is then extended to any Carrollian field $\sigma \in \mathcal{C}^{\infty}_{(\frac{3}{2},-\frac{1}{2})}(\mathscr{I}_{\mathcal{A}})$ by linear combinations, which proves the second equality of Proposition 1.

## Acknowledgments

Y.H. thanks Lionel Mason for several useful discussions. L.F. thanks Luca Ciambelli and Romain Ruzziconi for discussions. L.D. and Y.H. are grateful for the hospitality of Perimeter Institute where this work was initiated.

**Funding information**  This work is supported by the Simons Collaboration on Celestial Holography. L.D. is supported by the European Research Council (ERC) Project 101076737 – CeleBH. Views and opinions expressed are however those of the author only and do not necessarily reflect those of the European Union or the European Research Council. Neither the European Union nor the granting authority can be held responsible for them. L.D. is also partially supported by INFN Iniziativa Specifica ST&FI. Her research was also supported in part by the Simons Foundation through the Simons Foundation Emmy Noether Fellows Program at Perimeter Institute. Research at Perimeter Institute is supported in part by the Government of Canada through the Department of Innovation, Science and Economic Development and by the Province of Ontario through the Ministry of Colleges and Universities.

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
