# Peer review of "Carrollian $\mathscr Lw_{1+\infty}$ representation from twistor space"

_SciPost Physics, doi:SciPost Phys. 17, 118 (2024)_

## Round 1 · Referee Report · Anonymous (Referee 1) · 2024-9-4

Report

It has long been known that the self-dual sector of gravity in four spacetime dimensions is classically integrable, but only recently has this fact been connected with structures in perturbative gravitational scattering amplitudes. In particular, graviton wavefunctions on Minkowski spacetime can be written in a "conformal primary" basis, which (on a purely kinematical level) re-casts gravitational scattering amplitudes in the form of a conformal correlator on the two-dimensional celestial sphere. In this basis, there is an infinite tower of "conformally soft" gravitons, corresponding to residues of these conformal primary gravitons when their conformal dimensions take special (generically negative integer) values. It was then shown by Strominger that the set of conformally soft positive helicity gravitons forms an algebra under holomorphic collinear limits: $\mathcal{L}w^{\wedge}_{1+\infty}$, the loop algebra of the wedge algebra of $w_{1+\infty}$.

It was soon observed that the existence of this algebra had a simple explanation in terms of twistor theory, where it emerges as the algebra of infinitesimal deformations of a twistor space which preserve its underlying geometric structures. Furthermore, this twistor perspective makes it clear that a generic generator of $\mathcal{L}w^{\wedge}_{1+\infty}$ acts non-trivially on the radiative data at null infinity, $\mathscr{I}$, illustrating that the algebra is a symmetry of the self-dual sector rather than a particular self-dual metric. However, the explicit way in which this action is realized on the radiative degrees of freedom at $\mathscr{I}$ is not immediately clear from these twistor arguments, nor is the connection with another important realization of $\mathcal{L}w^{\wedge}_{1+\infty}$ on the gravitational phase space in terms of a tower of higher-spin charges.

This paper resolves these two important outstanding questions by explicitly tracing the action of $\mathcal{L}w^{\wedge}_{1+\infty}$ from twistor data to spacetime fields and their radiative degrees of freedom (i.e., the leading large $r$ coefficient in a Bondi-Sachs expansion near $\mathscr{I}$). This results in explicit, non-local expressions for the action of $\mathcal{L}w^{\wedge}_{1+\infty}$ on radiative degrees of freedom of both helicities, which precisely coincides with the canonical realization of the algebra in terms of higher-spin charges. They key step in this is a very non-trivial large $r$ expansion of the spacetime field to obtain the action on the radiative data (or Carrollian field) at $\mathscr{I}$.

This is an important result, which would merit publication in SciPost Physics in its own right, but in addition the paper provides an extremely clear and concise overview of both the twistor and canonical charge constructions. This makes it an excellent first port of call for researchers trying to learn about these ideas for the first time.

I confess to having very little in the way of critiques or suggestions for the authors; a few very minor typos are listed below which they may want to address prior to publication.

Requested changes

1) On page 2, "...the nonlinear gravitation construction..." should be "...the nonlinear graviton construction..."

2) On page 5, "We [de]note [by] $\mathcal{C}^{\infty}_{k,\bar{k}}(\mathscr{I})$ the space of Carrollian..."

3) On page 11, "...on a Carrollian fields..." should be "...on a Carrollian field..."

Recommendation

Publish (easily meets expectations and criteria for this Journal; among top 50%)

  • validity: top
  • significance: high
  • originality: high
  • clarity: top
  • formatting: excellent
  • grammar: good

Author:  Laura Donnay  on 2024-10-02  [id 4826]

(in reply to Report 1 on 2024-09-04)

We thank the Referee for their careful reading and positive comments on our manuscript.
We have corrected the typos pointed out.

---

## Round 1 · Referee Report · Anonymous (Referee 2) · 2024-9-6

Report

The appearance of the algebra Lw1+inf in celestial holography has generated great interest. It is related to the classical integrability of the self-dual sector of general relativity, whose basic consequence for graviton amplitudes is that these vanish at tree level, but there are also consequences beyond the self-dual sector itself. This paper explores the action of this algebra in two different formalisms, relating them explicitly. One formalism is that of canonical charges and asymptotic phase space, and the other is the twistor space formalism. The action of Lw1+inf has been studied in the two approaches independently, and the paper clarifies the map between the approaches.

This is a timely contribution that will be useful to what were previously two distinct communities working on these two formalisms. The paper is very well written. I recommend that it is accepted as it stands, unless the authors wish to make minor changes.

Recommendation

Publish (easily meets expectations and criteria for this Journal; among top 50%)

  • validity: -
  • significance: -
  • originality: -
  • clarity: -
  • formatting: -
  • grammar: -

Author:  Laura Donnay  on 2024-10-02  [id 4827]

(in reply to Report 2 on 2024-09-06)

We thank the Referee for their reading and positive feedback on our manuscript.

---

## Round 1 · Referee Report · Atul Sharma (Referee 3) · 2024-9-14

Strengths

1-This paper brings together different subfields working on celestial holography, namely the Carrollian approaches and twistor approaches.

2-It provides an explicit spacetime description of how twistorial w-infinity symmetries act on the fields of linearized gravity.

Weaknesses

1-Going from the twistorial to the Carrollian description increases the complexity of expressions by quite a lot, which can be quite impractical to use.

2-The paper works in a specific gauge where twistor data is obtained in terms of asymptotic shear, but does not present a potential path for making the analysis gauge invariant.

Report

This is a really good paper. It pulls together various subfields of the celestial holography program, including the ideas of asymptotic symmetries, aspects of gravitational phase space, and twistor theory. Until its appearance, the w-infinity symmetries of graviton scattering discovered by Guevara et.al. had been derived from various disconnected approaches. They could either be obtained from collinear limits of scattering amplitudes, from the analysis of asymptotic symmetries at null infinity, or from the twistor uplift of self-dual gravity. This paper is pivotal in proving the equivalence of all these approaches.

This paper meets the criteria of publication in SciPost Physics. I would be happy to recommend it for publication modulo the following suggestions for relatively minor changes.

Requested changes

1-In both figures 1 and 2, there are typos where the authors wrote $u=\mu\lambda$ instead of $u=\mu\bar\lambda$. Please correct these.

2-I am slightly unhappy with the authors dropping the inhomogeneous contribution $\bar\partial g$ in the gauge variation $\delta h = \bar\partial g + \{h,g\}$ of the twistor representative $h$. I think there might have been a slight conceptual misunderstanding. If one is perturbing around a nonzero "background" $h$, one needs to use the curved twistor lines that are holomorphic curves with respect to the complex structure $\bar\partial+\{h,\cdot\}$. When $\bar\partial g + \{h,g\}$ is pulled back to such a curved twistor line $L$, it simply becomes $\bar\partial|_Lg|_L$. When this is integrated over $L$, it picks out the singularities of $g(\mu,z)|_L$ in $z$, generating a nontrivial transformation of the background metric that one starts with. Could the authors explain why this is equivalent (maybe at linear order in $h$?) to just integrating $\{h,g\}$, and that too on the twistor lines of flat space? This also seems to be related to the authors worry about $\{h,g\}$ not giving a well-defined Penrose integral due to its singularities. I suspect they should instead use $\bar\partial|_Lg|_L$ as their integrand, which would be perfectly well-defined because it will just become a sum of derivatives of delta functions at $z=0$ and $z=\infty$.

In fact, if I perturb around a flat background $h=0$, then I would always get $\{h,g\}=0$ and hence no perturbation of the shear. This seems unsatisfactory because the action of w-infinity is supposed to add soft gravitons to the vacuum of flat space.

Recommendation

Ask for minor revision

  • validity: good
  • significance: high
  • originality: high
  • clarity: good
  • formatting: reasonable
  • grammar: perfect

Author:  Laura Donnay  on 2024-10-02  [id 4828]

(in reply to Report 3 by Atul Sharma on 2024-09-14)

We thank the Referee for his careful reading of the manuscript and interesting comments. We address the Referee's comments below.

1) We have corrected the typos in the figures. 2) We agree that the presence of the inhomogeneous term in the gauge variation is of physical interest as it captures the shift into different vacua. In this paper, we wished to derive and highlight the representation of the w-infinity algebra on Carrollian fields (hence the linear, as opposed to affine, action). To answer the Referee's question, we do not think that this is equivalent to linearizing the action of w-infinity on the complex structure, which would involve both homogeneous and inhomogeneous terms. As pointed out by the Referee, the 'soft' (inhomogeneous) action on the Carrollian fields can be seen to localize and be expressed in terms of delta-function on the celestial sphere at $z=0$ and $z=\infty$.

---

## Round 2 · Referee Report · Atul Sharma (Referee 3) · 2024-10-7

Report
The authors have answered my questions satisfactorily and I'm happy with the corrections made. I recommend that this paper be published in SciPost Physics.
Recommendation
Publish (meets expectations and criteria for this Journal)

---

## Editorial Decision

published